# Comprehensive Review of Power Electronic Converters in Electric Vehicle Applications

Rejaul Islam [1] , S M Sajjad Hossain Rafin [2] and Osama A. Mohammed [2,*]

1   EEE Department, Bangladesh University of Engineering and Technology, Dhaka 1205, Bangladesh
2   Energy Systems Research Laboratory, ECE Department, Florida International University, Miami, FL 33174, USA
*   Correspondence: mohammed@fiu.edu

**Abstract:** Emerging electric vehicle (EV) technology requires high-voltage energy storage systems, efficient electric motors, electrified power trains, and power converters. If we consider forecasts for EV demand and driving applications, this article comprehensively reviewed power converter topologies, control schemes, output power, reliability, losses, switching frequency, operations, charging systems, advantages, and disadvantages. This article is intended to help engineers and researchers forecast typical recharging/discharging durations, the lifetime of energy storage with the help of control systems and machine learning, and the performance probability of using AlGaN/GaN heterojunction-based high-electron-mobility transistors (HEMTs) in EV systems. The analysis of this extensive review paper suggests that the Vienna rectifier provides significant performance among all AC–DC rectifier converters. Moreover, the multi-device interleaved DC–DC boost converter is best suited for the DC–DC conversion stage. Among DC–AC converters, the third harmonic injected seven-level inverter is found to be one of the best in EV driving. Furthermore, the utilization of multi-level inverters can terminate the requirement of the intermediate DC–DC converter. In addition, the current status, opportunities, challenges, and applications of wireless power transfer in hybrid and all-electric vehicles were also discussed in this paper. Moreover, the adoption of wide bandgap semiconductors was considered. Because of their higher power density, breakdown voltage, and switching frequency characteristics, a light yet efficient power converter design can be achieved for EVs. Finally, the article's intent was to provide a reference for engineers and researchers in the automobile industry for forecasting calculations.

**Keywords:** transportation electrification; electric vehicles; power converters; third harmonic injection; multi-level inverter

## 1. Introduction

A rapidly growing market of electric vehicles (EVs) has been witnessed in the last decade since the vehicles are environmentally friendly, have enough resources, and are a counterpart of gasoline-powered internal combustion engine (ICE) automobiles. In 2020, the transportation sector caused 27% of total GHG emissions in the US [1,2]. Hence, one of the best reasons behind considering electrified automobiles over conventional automobiles is that they eliminate many issues, such as greenhouse gases (GHG) caused by traditional automobiles [1]. Moreover, electrified cars can increase efficiency, acceleration, and overall performance and eliminate harmful GHG emissions and maintenance costs [1,3]. A comparison between EVs and ICE vehicles has been made in [4]. The comparison was in terms of the time required for 845 km inter-city journey. It was disclosed that based on the present battery capabilities, power charges of more than 400 kW are required to achieve a comparable travel time between EVs and ICEVs. Moreover, the possible solution to this matter is utilizing high-speed chargers (XFCs) with the capability of providing at least 800 V direct current (DC) at the output [5].

The whole power supply system of an EV can be divided into three parts, namely: (i) battery charging system, (ii) powertrain, and (iii) regenerative braking. The block diagram of a typical EV with its power supply system is depicted in Figure 1. This figureshows that a particular DC–DC converter may be required for an individual energy source to be integrated into the high voltage (HV)-DC bus of the EVs and PHEVs powertrain [6].

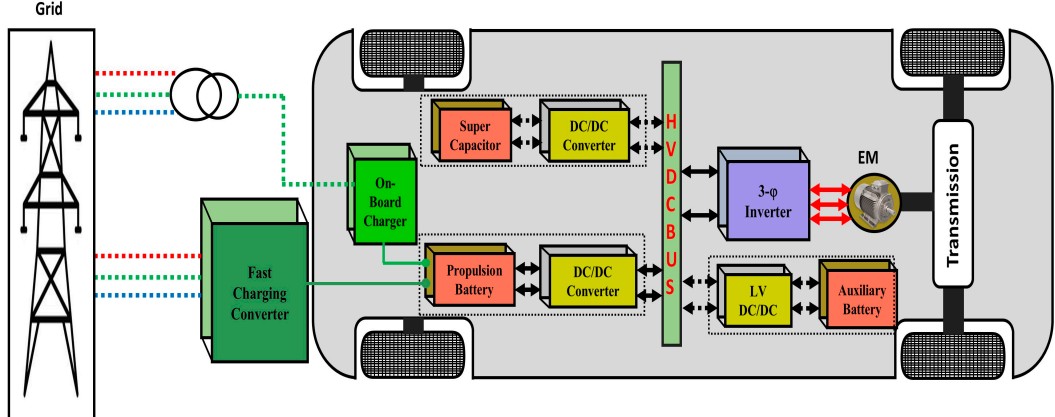

**Figure 1.** Schematic diagram of an EV powertrain with fast-charging stations [6].

In a battery charging system, all the input sources, such as batteries and supercapacitors, are charged via an AC–DC rectifier from the charging three-phase AC grid. Hence, the battery charging system of electric vehicles plays a critical role in developing EVs. The charging system of an EV is typically known as conductive charging, which is generally categorized as an onboard and offboard charging system. Onboard chargers are installed inside the EV; contrary offboard chargers are established outside the EVs [7]. A more extensive discussion on EV charging systems will be presented in Section 2.

In the EV powertrain, all the electric input sources are connected to a HV–DC bus by an individual DC–DC converter, and the output three-phase electric motor (EM), i.e., the main load of the EV, is powered from this HV–DC bus through a three-phase inverter which drives the EM [8–11]. Here, the voltage level of this HV–DC bus of the EV is around 400–750 V. Moreover, moving the EV via an EM from electric batteries, a DC–DC voltage is required because the batteries' output voltage is much lower than the required voltage of EM. A traction inverter is needed to drive the EM by converting the DC batteries into variable-frequency AC [12]. However, a disagreement could be made for stepping up the output AC voltage level of the inverter by utilizing a high-voltage transformer instead of a DC–DC converter. This is due to its having several essential advantages, such as reliability, cost-effectiveness, compact size, and lightweight DC–DC converters appear to be excellent candidates for EVs and HEVs powertrains [13]. An extensive discussion on the EV powertrain converter topologies is presented in Sections 3 and 5. Finally, regenerative braking, capable of charging EV batteries between driving-braking times by capturing electricity, was discussed.

The comparative performance analysis of different electric motors, such as permanent magnet (PM), induction motor (IM), switched reluctance machines (SRMs), and synchronous reluctance machines (SyncRels) were presented in this review paper. The SRM is considered a strong candidate for EVs due to its robust structure and low cost [14]. Moreover, the structural integration of the electric motor drive was briefly discussed in this paper to increase the overall system's efficiency, with installation and manufacturing cost reductions of between 30 to 40 percent [15].

Although there are many benefits of using electrified vehicles (EVs) instead of conventional automobiles, the long charging times, queuing times at charging stations, and range anxiety are of concern. The latter is due to the available battery technology and related issues such as energy density, cost, and capacity, which are the main challenges blocking

the further development of EVs [16–18]. Therefore, the main limiting factor of the fast reception of EVs is the unavailability of efficient charging infrastructure compared to ICE vehicle refueling stations [19].

The limitation of the charging infrastructure of EVs can easily be overcome by utilizing extremely fast chargers and wireless power transfer (WPT) technology since the vehicle will be charging while in motion or at rest [7]. The fast charger is able to recharge an EV battery from empty to 80% within 20–25 min. As a result, it will reduce the queuing time at the charging stations. Also, the limitation of the charging infrastructure of EVs can easily be overcome with wireless charging technology since the vehicle will be charging while at rest or in motion. This article intends to help engineers, designers, and researchers in the EV field to forecast and choose the best chargers for fast charging and gather informative knowledge on wireless charging. The design parameters of the EV power electric charger were discussed, and the parameters were verified with MATLAB/Simulink simulations. The main objective of this article was to forecast the future of wireless charging in EV field to mitigate the currently available limiting factors.

This automatic WPT charging can be achieved with two basic resonant power transfers: the resonant inductive power transfer (IPT) and the resonant capacitive power transfer (CPT). Both IPT and CPT operate with three different charging modes: (i) static wireless charging (SWC); (ii) quasi-dynamic wireless charging (QWC); and (iii) dynamic wireless charging (DWC) [20]. One of the best advantages of using SWC is that it can be installed in convenient locations, such as home garages or parking lots, and can eliminate the shock risk [21]. On the other hand, EVs can be charged with QWC systems while they are stopped for a short time, such as at traffic lights. Finally, the DWC system is more famous due to its continuous charging capability since it can charge vehicles en route [22]. As a result, it increases the driving range of electric vehicle powertrains while decreasing the battery size. However, EVs' dynamic wireless charging technology is still in its embryonic stage. To implement the vision of wirelessly powered electric vehicles, numerous challenges related to safety, efficiency, effective power transfer, the distance between coils and plates, misalignment, cost, and performance must be overcome. A more extensive discussion of the WPT system is presented in Section 2.

Furthermore, the demerits of the charging power converters, wireless power transfer, energy storages, and electric motor can be overcome by utilizing wide bandgap semiconductors (WBGSs), such as gallium nitride (GaN) and silicon carbide (SiC) based devices [23,24]. Numerous advantages, including higher switching frequencies, higher temperature operation, lower losses, etc., can be achieved by WBGS-based power converters, such as rectifiers, and DC–DC converters and inverters, over silicon semiconductor device-based power converters [12]. In [25–31], several wireless charging systems with 50 kW, 60 kW, 100 kW, 200 kW, 250 kW, and 500 kW power levels were published for electric vehicles, buses, fleets, and trucks with and without wide bandgap (WBG) semiconductor devices. They have also depicted that the efficiency of these wireless charging systems remains around 91–97%. In motor drives, several advantages, such as low switching and conduction losses, high power density, lower ON-state resistance, and high-temperature operation, can be achieved by utilizing wide bandgap semiconductor devices [32,33]. Moreover, this review paper discusses the comparison among GaN, SiC, and Si with the current status, challenges, and different WBG semiconductors trends. The comparison showed that the WBGSs provide tremendous advantages, such as faster switching frequency, lower loss, and higher efficiency over silicon semiconductors [15]. Due to these characteristics and the commercialization progress of silicon carbide (SiC) and gallium nitride, they are considered the most promising WBGS nowadays [34].

Some of the most recent overview literature papers on power electronic converters (i.e., AC–DC, DC–DC, DC–AC), suitable electric motors, energy storage, wireless charging, and utilization of wide bandgap semiconductor devices for EV applications are enlisted in the following Table 1. These literature summaries will help the readers to gain more knowledge in every section of this paper.

**Table 1.** Recent overviewed literature on power electronic converters, electric motors, and energy storage for electric vehicle applications.

| EVs Power Electronic Converters Overview Contributions | Recently Overviewed Articles | Citation | Addressed in this Overview |
|---|---|---|---|
| Overview of charging rectifiers | [5,6] | 104;198 | √ |
| Overview of powertrain DC–DC converters | [6,35] | 198;1339 | √ |
| Overview of powertrain multilevel inverters | [1,12,36,37] | 73;164;269;39 | √ |
| Overview of electric motors for EV | [15,38,39] | 92;66;11 | √ |
| Overview of energy storage for EV | [40] | 677 | √ |
| Overview of wireless power charging | [7,16,41,42] | 518;97;551;30 | √ |
| Overview of the utilization of GaN and SiC | [32–34,43] | 14;25;1874;72 | √ |

As mentioned, in recent years, significant progress has been made in the research and commercial development of electric vehicle applications because of their rapid growth. Because electric vehicles (EVs) are a unique mobile source of demand, EV power and control are different from typical residential, commercial, and industrial loads. Furthermore, this emerging electric vehicle (EV) technology requires high-voltage energy storage systems, efficient electric motors, electrified powertrain, and power converters. Evidently, EV technology demands the use of forecasting in many aspects of EV fields because of its dynamic nature.

In a typical scenario, among any other household appliances, EV charging requires more power and needs a relatively long charging time. As a result, EVs place significant coincident demand patterns and volume on the power system, specifically in the distribution system. Moreover, EVs may introduce the system's peak demand, exceeding voltage limits, and overloading lines, and transformers [44]. For the system to prevent faults in the system and other stability issues, knowledge of day-ahead demand is crucial. Thus, to ensure uninterrupted power supply, forecasting and analyzing the impact of the uncertain demands caused by electric vehicles on the power system. Having a stochastic nature, the EV load demand is heavily influenced by the driving and traveling patterns of each EV owner.

Similar to the uncertain charging demand increase, driving an EV itself demand forecasts. The movement of objects' current and future locations in the driving scene is uncertain. Furthermore, control design for autonomous cruise control, emergency braking, and lane-keeping assistance systems is a real challenge in cases such as high-speed driving on a highway, deadlock situations, and/or rough driving by other drivers [45]. Forecasting and predicting EV charging demand and driving conditions are calculated using conventional statistical methods and artificial intelligence [46]. Classical or contemporary control schemes could not work properly for autonomous driving in case of uncertain weather conditions, the presence of the state, input constraints, and modeling errors considering the nonlinear dynamics of the vehicle [47].

Conventional statistical methods and artificial intelligence are used for calculating forecasts considering all these parameters mentioned above. However, the impact of power electronic converters, control devices, methods, and energy storage are somewhat neglected. In EV applications, numerous types of converter topologies and their dynamic control schemes are used from the charging point to the transmission, which can be seen in Figure 1.

Moreover, onboard and/or offboard chargers that are utilized are high-power converters, where AC to DC and DC to DC power conversion occurs. These converters are the interface that channels required power to be stored in batteries to run an EV. Thus, information regarding charging characteristics of the converter topology, control method, thermal attributes, switching losses, and efficiency is vital for the EV to work properly, which is, in fact, directly related to the power system demand scenario discussed in the pre-

vious paragraph. Significantly, characteristics of the battery management converters could also help the prediction calculation of the charging and discharging time. Furthermore, to perfectly predict the demand for near or far forecasts, the type of charging converter, control method, and speed could be utilized.

While driving an EV, whether it is self-driven or autonomous, reacting quickly during uncertain conditions could avoid a fatal accident. This is where forecasting, i.e., predictive control, plays a crucial role in cruise control, emergency braking, and lane-keeping assistance systems. Importantly, forecast calculations of such events could utilize current information of various components which give power and run the EV, such as battery management converters, motor drive, and their controls. All these power components work 100 percent during run conditions. Thus, the present data of these components are vital for future actions. While predicting the forecast, information on these converters' characteristics, control method, thermal attributes, switching losses, and efficiency is needed could potentially be vital in every aspect.

Therefore, this paper aims to value the importance of power electronic converters and their control in the forecast of electric vehicle applications. It also aims to investigate the power electronic converters and their control schemes in EV applications, particularly forecast calculation. Moreover, to compare them with the available ICE-based vehicles in terms of safety, environmental friendliness, efficiency, gain, and cruising range. After this brief introduction, a detailed review of the EV charging system is presented in Section 2. Comparative performance analysis of different types of DC–DC converters for EV applications has been conducted in Section 3. A detailed discussion of the opportunities and challenges of higher voltage energy storage for EVs is presented in Section 4. Section 5 compares two-level inverters (TLI) and multi-level inverters (MLI) in terms of efficiency, power density, performance, cost, etc., for driving an electric vehicle via EM. Section 6 presents an elaborate discussion of several electric motors with their torque, dynamic response, speeds, cost, fault tolerance, etc., is presented. Section 7 analyzes all types of power electronic converters' simulation results and discusses the best-suited converter topologies for EVs and HEVs. Section 8 presents a comparative functionality analysis and summaries of all power electric converter topologies. In Section 9, future trends of EV research with WBGSs and system integration are reviewed extensively. Finally, Section 10 concludes by commenting on the best-suited power electronic rectifiers, DC–DC converters, inverter topologies, wireless power transfer technologies, energy storages, and electric motors for EVs and HEVs.

## 2. Charging Section

The charging system of electric vehicles plays a critical role in the development of EVs [7], which is done by utilizing a $3\varphi$ AC–DC active rectifier. In the modern EV industry, $3\varphi$ active rectifiers are becoming more popular than passive bridge rectifiers due to their ability to increase the electromagnetic properties and recover electric motor energy to the supply network [48–52].

The battery charging mode is categorized into slow and fast charging modes based on the charging power level of the chargers. The slow-charging mode uses an alternating current and has a charging power of about 1.5 kW. As a result, charging EVs directly from domestic power systems via slow chargers is certainly advantageous. The issue is that charging a 24 kWh EV battery from empty to full will take approximately 16–17 h at a charging power of 1.5 kWh, which requires the EV owners set aside enough time for charging. On the other hand, the fast-charging mode employs DC rather than AC. To obtain a comparatively short charging time, fast chargers deliver power at up to 50 kW or more. Tesla constructed 120 kW supercharger stations for fast charging and with this supercharger a 90 kWh battery of the Model S can be charged from empty to 80% in about 40 min. Because of extremely high currents and voltages, these fast-charging opportunities are more high-tech than a simple residential outlet [53].

Moreover, to refill/recharge the energy storage devices, chargers are required, and in terms of mobile platforms, they can be classified into two categories: (a) conductive charging, and (b) wireless charging [7,54].

### 2.1. Conductive Charging for Electric Vehicles

A physical connection between the power supply and the electric vehicle is involved in a conductive charging system, which consists of two main parts: (i) AC–DC (rectifier) converter, and (ii) a DC–DC converter. There are two types of conductive chargers available in the EV market: (1) onboard chargers, and (2) offboard chargers. Onboard chargers are installed inside of the electric vehicle. On the contrary, offboard chargers, like refueling stations, are established outside the EVs [7].

### 2.1.1. Onboard Charging

Onboard charging is the best-suited solution for the regular charging of EV users [49]. A conductive charging system involves a physical connection between the power supply and the electric vehicle [7]. An onboard charger reduces most of the time by providing independent charging features because it can be directly connected to the AC grid, providing a universal charging station solution. This is since EVs can be charged in any residential area with 240 V voltage and a 30 A current rating [54]. In the following sections two popular topologies such as (a) phase shift modulation topology and (b) Full-bridge LLC resonant converters, are discussed, which are usually utilized as onboard chargers. To provide isolation between the vehicle and the grid, a high-frequency transformer is utilized with a series LC/LLC resonant topology and a phase-shift modulation topology in onboard chargers [54].

Phase Shift Modulation Topology

The most popular charging converter in high-power appliances is the isolated phase-shifting-based full-bridge converter (PSFBC), illustrated in Figure 2. A phase-shifted full-bridge converter provides various desirable advantages over other charging converters, such as simple control techniques, low current stress on devices, etc. [6]. Moreover, the phase-shift modulation topology provides an alternative solution for a wide-range charging operation [55–57]. The isolated PSFBC has two operation stages: the DC–AC converter (inverter) stage with a high-frequency transformer, and the AC–DC rectification stage. As shown in Figure 2, both stages consist of four power-switching elements to make a full-bridge like a conventional full-bridge converter. The isolated phase-shifted full-bridge converter has ten working modes which are discussed in [58].

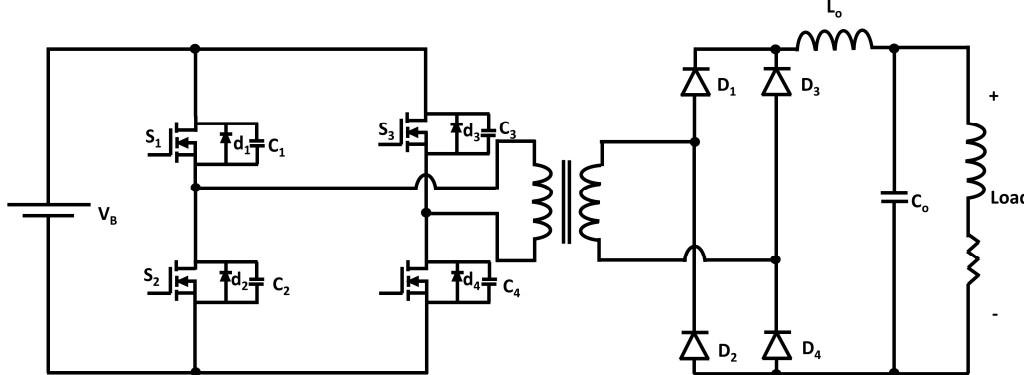

**Figure 2.** Isolated full-bridge phase shift converter.

Despite many desirable features, the converter has some drawbacks, such as the rectifier bridge facing high voltage stress and high-circulating current in the freewheeling interval [6]. Moreover, to improve the efficiency, volume, cost-effectiveness, and reliability

of this onboard charging topology, researchers have been investigating several possibilities, such as phase-shifted dual-active-bridge (PSDAB) [59,60] and phase-shifted triple-active-bridge (PSTAB) [61].

To charge electric vehicles, a particular voltage level is required; otherwise, the battery can be damaged, or its lifespan can be reduced. A control system, namely a servo system, is capable of regulating controlled variables to their reference inputs without steady-state errors against unknown disturbances or voltage levels [58]. In the design process of a control system, there are two critical issues: compensation of certain types of deterministic disturbance, and particular reference signals [62].

A closed-loop control scheme is depicted in Figure 3, which consists of a controller C(s), disturbance D(s) with input U(s), plant Go(s), reference R(s), Error E(s), and Output Y(s) [63].

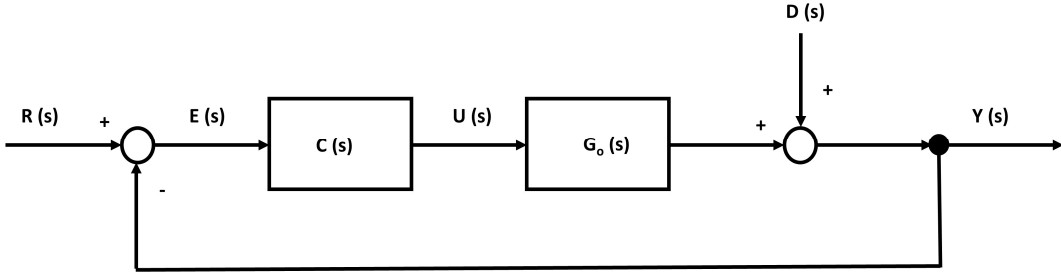

**Figure 3.** Basic block diagram of a closed-loop control system [63].

The error signal E(s), which is the difference between the output and reference signal, can be expressed as [63],

$$E(s) = \frac{1}{1 + C(s)G_o(s)} \left( R(s) - D(s) \right)$$

There are several control schemes presents for controlling AC–DC rectifiers, such as proportional (P) controller scheme, integral (I) controller scheme, proportional-integral (PI) controller scheme, proportional-integral-derivative (PID) controller scheme, pulse width modulation (PWM) scheme, Sinusoidal PWM, third harmonic injection (THD) PWM scheme, space vector pulse width modulation (SVPWM) scheme to name a few.

Moreover, conventional PI controllers are generally utilized to control the output voltage of the phase-shifted full-bridge converter due to several drawbacks, such as undesirable dynamic characteristics and the frailty to disturbances of the converter's several control schemes, e.g., the PI gain-scheduling controller [64], hybrid fuzzy logic sliding mode controller (HSMC) [65], an optimal dead-time control scheme with burst-mode operation [66], Laguerre function-based model predictive control (MPC) [67] have been proposed, designed, and presented in [64–67]. Due to the conventional PWM counterpart of the PI gain-scheduling controller, the continuous and adjustable voltage range of 0–50 V can easily be achieved, which is experimentally justified in [64]. Implementing the A/D (analog to digital) converter with a field-programmable gate array (FPGA)-IC makes the controller compact in size and achieves flexibility. The hybrid fuzzy logic sliding mode controller can improve the dynamic characteristics, stability, and robustness from disturbance; and can easily mitigate the chattering demerits, output current, and voltage ripples of the PSFBC by varying the SMC gain [65]. An optimized dead-time for the PSFB converter has been proposed and experimentally verified in [66], which provides the natural burst-mode operation. As a result, excellent thermal performance and low power consumption can be achieved at both heavy and light loads. The Laguerre function-based model predictive control (MPC) scheme is an excellent candidate for PSFBC due to its several compelling advantages, such as non-linear peak input current constraint and multiple physical constraints compared to the conventional PI-based controllers [68]. Furthermore, the proposed

controller has been experimentally tested andvalidated that the Laguerre function-based model predictive control (MPC) scheme is an effective control scheme for the isolated phase-shifted full-bridge converter [67].

Isolated Full-Bridge LLC Resonant Converters

In the EV charging stage, another powerful converter, the full-bridge LLC resonant converter, can be used, which can achieve maximum efficiency as an EV charger [54,69–72]. The converter is depicted in Figure 4, which can perform a full range of zero voltage switching for the power switching devices and high efficiency at high voltage operation. Moreover, there is no reverse recovery current, and the rectifier diodes do not face any oscillation voltage due to the sinusoidal current waveform of the FB-LLC shape [6]. The isolated FB-LLC converter has seven operational modes with five working stages, which are discussed in [73].

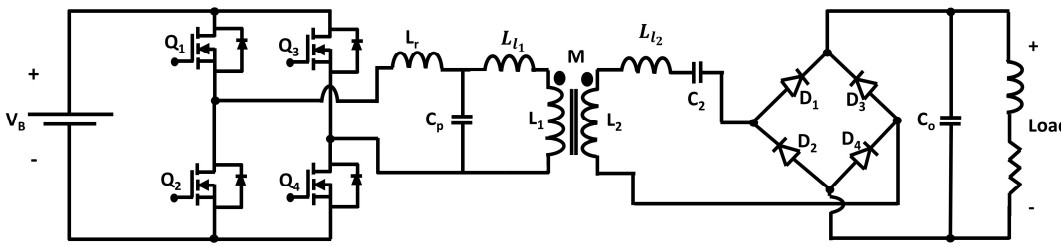

**Figure 4.** Isolated full-bridge LLC resonant converter.

Despite many merits, this converter topology needs a high switching frequency range to control the battery's output voltage, which is this converter's major demerit. Due to this reason, the efficiency of this converter becomes less when the battery voltage becomes low, and the design of the filter and transformer becomes much more complicated. Nevertheless, to improve the efficiency, volume, cost, and reliability of this onboard charging topology, researchers have been investigating several possibilities: resonant LC half-bridge with the front-end PFC [56]; bidirectional LC resonant converter with a full-bridge front-end interface [57]; LLC with a SEPIC front-end PFC [74]; and LC-LLC bidirectional converter with an interleaved output stage [75].

For the isolated full-bridge LLC converter topology, enormous new control schemes, such as the frequency adaptive phase-shift modulation control scheme [76]; secondary phase-shift control scheme [77]; fixed-frequency PWM control scheme [78,79]; hybrid control strategy of phase-shift and variable-frequency control scheme [80]; simplified optimal trajectory control (SOTC) scheme [81,82]; digital direct phase-shift control (DDPSC) scheme [73]; and resonant frequency tracking method [83] etc. have been proposed and designed. The designs achieve wide-range input voltage, accurate shutdown time settings, dynamic response, and load mutations rapid adjustment [76–83]. Among them, the digital direct phase-shift control (DDPSC) scheme shows excellent performance in controlling FB-LLC converters due to its simple design and flexible structure, improved dynamic response, soft switching over a wide load range, and constant voltage output when the gain is less than 1, which were verified via experimental results in [73].

2.1.2. Offboard Charging

Offboard charging can reduce the impact of EVs on the grid with proper planning and managing peak demand [54]. A fast charger is required to mitigate the refueling problem, which requires more time for refueling/recharging, fewer charging stations, and a lower driving range [7]. The fast-charging architectures are usually classified into two categories: (i) common AC-bus architecture; and (ii) common DC-bus architecture [84,85]. In the typical AC-bus architecture, each fast charger consists of individual AC–DC conversion stages with a high-frequency transformer, as depicted in Figure 5. On the other hand,

the typical DC-bus architecture consists of one central AC–DC conversion stage with a low-frequency transformer, which is shown in Figure 6 [6].

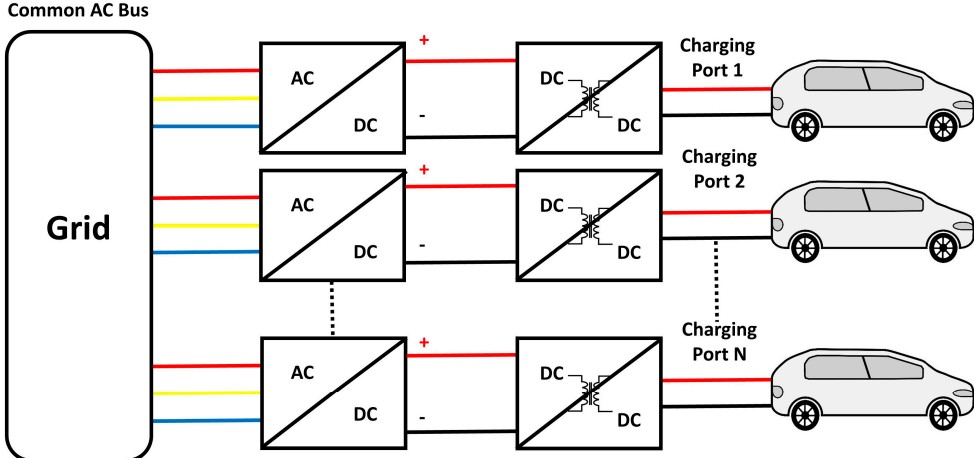

**Figure 5.** The charging station of BEVs and PHEVs with common AC bus architecture [6].

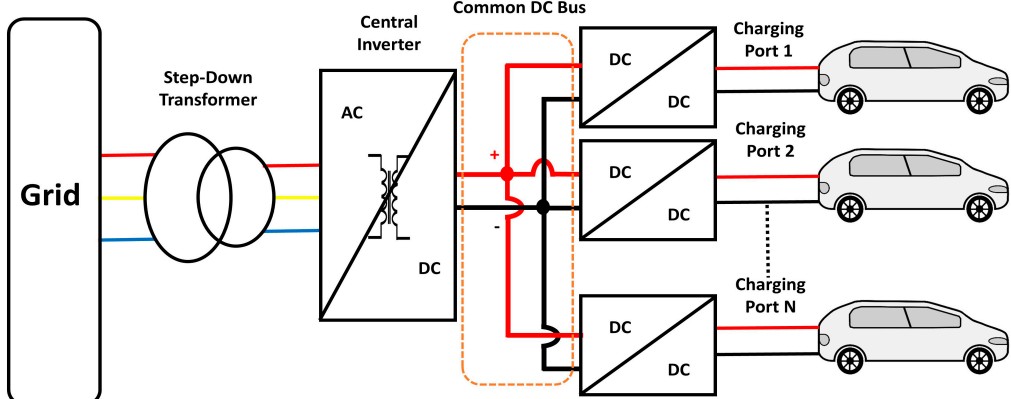

**Figure 6.** The charging station of BEVs and PHEVs with common DC bus architecture [6].

Furthermore, in [6], several types of AC–DC converter topologies for fast charging have been investigated. Among them, (a) three-phase bridgeless boost rectifiers and (b) three-phase Vienna rectifiers are commonly utilized in offboard charging stations..

Three-Phase Bridgeless Boost Rectifier

In EV applications, three-phase bridgeless boost rectifiers are conventionally utilized as the interface between the charging section of EV and the three-phase power supply of the grid due to their capability of converting three-phase AC into DC [86,87]. Among several types of PWM rectifier topologies, three-phase voltage source PWM rectifiers (i.e., three-phase bridgeless boost rectifiers) are widely utilized in electric vehicle applications and power generation due to their good voltage regulation, high efficiency, small input current, and output voltage filters size, and fast switching [88–93]. Hence, low-frequency harmonics can be quickly suppressed due to the high switching frequency. The unity power factor can be obtained with the most popular pulse-width modulation (PWM) rectifier topology [88,92]. Figure 7 shows the three-phase bridgeless boost rectifier, which takes advantage of the power factor correction (PFC) boost converter [6]. As the name suggests, this rectifier can boost or step up the output DC voltage and is capable of bidirectional power flow with the regenerative braking of the electric vehicle, which increases the battery efficiency [92].

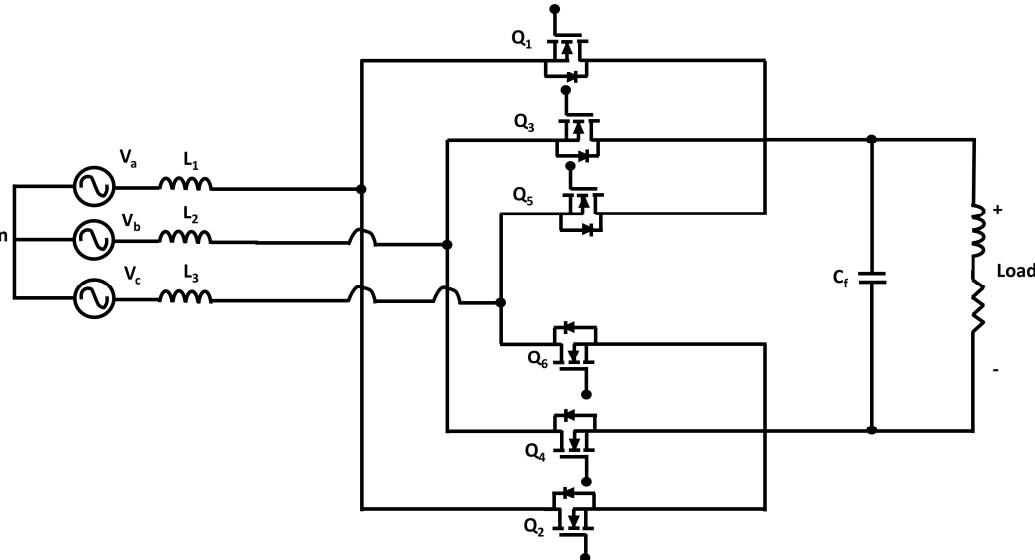

**Figure 7.** Three-phase bridgeless boost rectifier.

Each phase of the three-phase bridgeless rectifier consists of one coupled inductor, which serves as the ac-side filter, and two switches. The parallel capacitor Cf at the output side acts as the DC-side filter, which filters out the output voltage ripples [87]. There are six modes in the three-phase bridgeless boost converter operation, and during each mode, two switching devices are conducted. The switching sequences of the switches determine the charging and discharging of the inductors L1, L2, and L3 [6].

Furthermore, the three-phase boost rectifier provides controllable DC-bus voltage and power factor, low harmonics, high performance, high efficiency, bidirectional power flow, and fast dynamic response [94]. However, due to electromagnetic interference (EMI) noise, a harmful shoot-through state is caused in the rectifier, which makes the three-phase bridgeless boost rectifier vulnerable [95].

The three-phase bridgeless boost rectifier has enormous control processes such as space vector modulation (SVM), sliding mode control (SMC), hysteresis current control (HCC), and sinusoidal pulse-width modulation (SPWM), which are discussed in [96–99]. Some of these papers are focused on minimizing the stresses on active devices. Still, most of these papers favor reducing the filter size (i.e., inductor and capacitor), and high-power stress is reported for all these control processes due to the nature of the boost converter [6]. In [63], a new PWM control technique, namely a digital repetitive control scheme, was proposed to achieve zero tracking error for a three-phase PWM AC–DC converter. It is a conventional feedback controller developed systematically with complete robustness and stability.

Furthermore, to handle the measurement errors in the three-phase PWM rectifier, a compensation approach was introduced in [100], where they suggested a compensation method to mitigate the input three-phase imbalanced currents and output DC voltage ripples, which were created due to the resizing measurement errors and the DC compensation. In [51,101], for controlling power converters, finite-control set-dependent model predictive control (FCS-MPC), and space vector modulation-based deadbeat control (DBC) schemes were investigated. It was discussed that the tracking error might be reduced by combining both predictive techniques into a standard structure.

Although the SMC has a variable switching frequency issue, the fast-dynamic response, robustness, and insensitivity to system parameter variation suitable ways to control the rectifier [102–106]. The control scheme is one of the must-have components for controlling electric vehicle power converters and for constant output voltage levels to mitigate system failure or/and damage.

Vienna Rectifier

Three-phase bridgeless rectifiers utilize pre-regulators to control the power factor (PF) and harmonics of the input current drawn from the utility power supply. These pre-regulators serve as a resistive load on the power supply, and they are essentially non-linear converters with switching controlled in such a way that they function like a linear load with unity PF. The addition of a pre-regulator maximizes circuit complexity, system cost and contributes to additional power loss. This can be mitigated by using a simple, low-cost, high-efficiency pre-regulator circuit topology with a high input PF and low current harmonics. Having taken the aforementioned aspects into account, the VIENNA rectifier has emerged as the obvious choice as a pre-regulator for three-phase high power applications due to its simple architecture, low active switches, and high efficiency [107]. Figure 8 depicts the circuit diagram of a Vienna rectifier. Moreover, the Vienna rectifier has fewer active switching devices than the three-phase bridgeless converter [108], which reduces the reverse recovery current losses and makes the converter compatible with fast charging. Hence, due to several advantages such as simple construction, high power, very few active switches, high efficiency, low cost, pre-regulator for three-phase, and low current harmonics, this rectifier is widely utilized in the field of electric vehicle power converters [107].

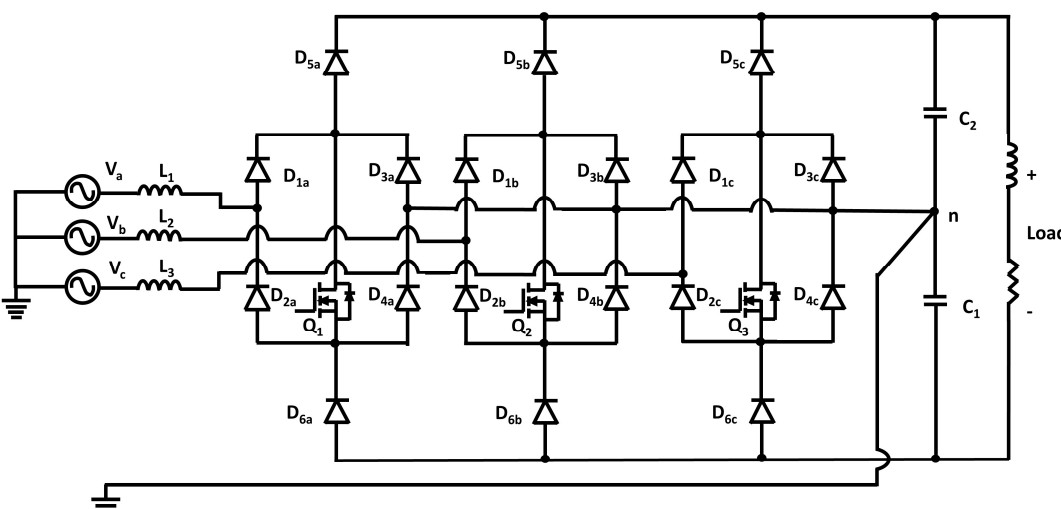

**Figure 8.** Three-phase Vienna rectifier.

The Vienna works mainly in two operational cycles, the positive and the negative cycles. Both cycles have two working modes. Therefore, a three-phase Vienna rectifier has four operating modes, briefly discussed in [107].

Due to the fast-charging control algorithm, several signal processors and microcontrollers are needed, which increases the chargers' overall cost and the battery's degradation [7]. Nevertheless, a high voltage is required on the output side of the three-phase Vienna rectifier [6].

The interest in Vienna rectifiers has been continuously increasing in academic and industry fields. As a result, researchers are increasingly interested in the control scheme and topology of the Vienna rectifier [109–115]. To control the rectifier, the sign of the input voltage and the rectifier has two dc-link capacitors. These two main characteristics must be considered [116–118]. Moreover, the model predictive control (MPC) methods have been proposed recently to control the rectifier better and more efficiently [112,113,119]. In [120], a discrete space-vector modulation (DSVM) with a finite control set-MPC (FCS-MPC) for the Vienna rectifier was proposed and analyzed that the converter can achieve high performance and efficiency for both the low current ripples and the fast-dynamic

response. In [116,117], a carrier-based SVM was proposed for Vienna rectifiers, in which PI controls schemes can also be utilized.

### 2.2. Wireless Charging

As discussed earlier, long charging times and lower operating times are significant challenges in EVs. To overcome these challenges, EVs require high-speed charging (XFC) and more battery capacities [54]. Hence, adding more battery cells and modules to improve the cruising range of EVs also increases the cost and weight of the vehicle [121]. Therefore, a flexible, extensive, and efficient charging solution is required to solve the range anxiety without increasing the battery cells and modules. Wireless power transfer (WPT) technology, first proposed by scientist Nikola Tesla, can be an excellent solution to mitigate this problem [16]. WPT technology transfers energy via magnetic field couplers, hence it eliminates the excessive connectors and charging cords and offers high operational flexibility, is vandalism proof, and weather resistant. As a result, electric vehicles with wireless charging can be charged while driving anywhere, and at any time [122].

2.2.1. Classification of Wireless Power Transfer (WPT)

The WPT can be divided into three main categories according to the power accumulation and transfer medium: (a) microwave radiation type (far-field) WPT; (b) electric field (capacitive) coupled type WPT; and (c) magnetic field (inductive) coupled-type WPT [41,123,124]. The classification of the wireless power transfer system of the electric vehicle is illustrated in Figure 9.

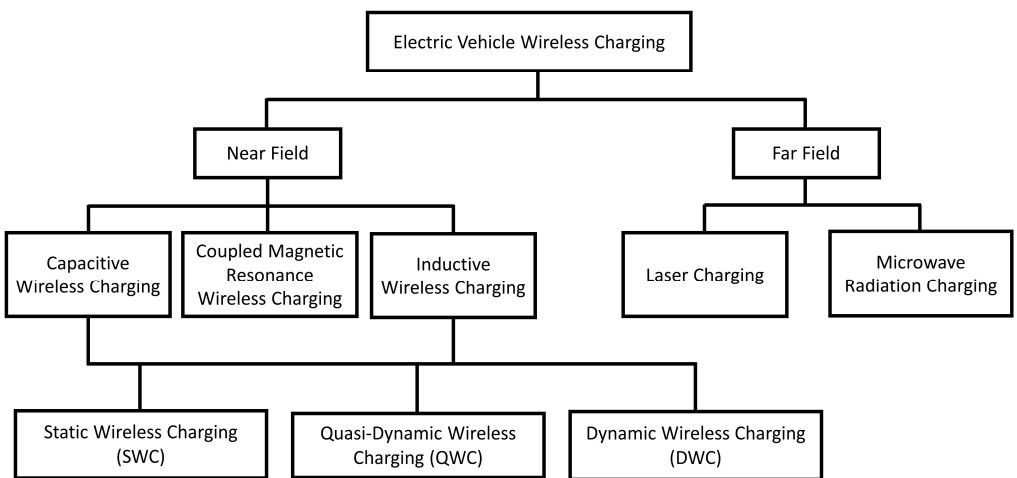

**Figure 9.** General classification of EVs wireless charging system.

Microwave Radiation WPT System

The microwave radiation wireless power transfer system is mainly used in long-distance power transfer, which is depicted in Figure 10, where the energy transfers to the output through DC/RF conversion, transmitting antenna, receiving antenna, low-pass filter, matching network, and rectifier [16].

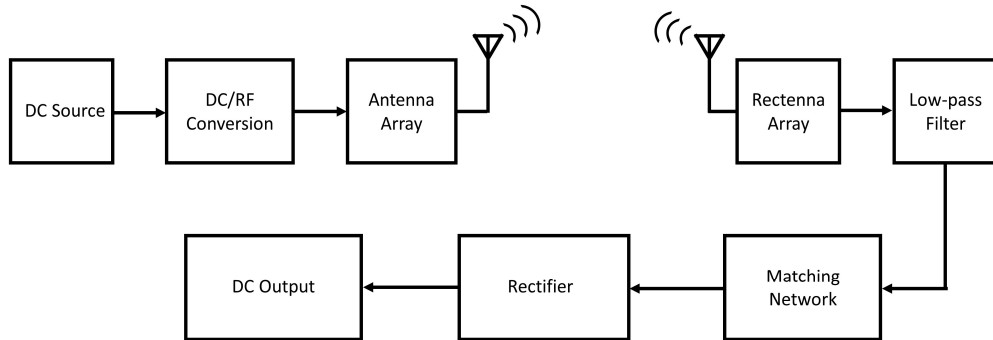

**Figure 10.** Diagram of WPT based on microwave technology [16].

Capacitive WPT System

The near-field electric field-coupled WPT system, also known as the capacitive power transfer (CPT) system, is depicted in Figure 11, where two pairs of plates are utilized to transfer power [16]. The transmitting section of capacitive power transfer (CPT) consists of the power grid, rectifier, high-frequency inverter, and primary capacitive plate. On the other hand, the receiving section of CPT, which is incorporated in the EV, consists of a secondary capacitive plate, rectifier, and battery [42]. The primary capacitive plate, including the transmitting section, is buried underground in traffic lights or parking slots, and the EV keeps the secondary or receiving capacitive plate [16,42].

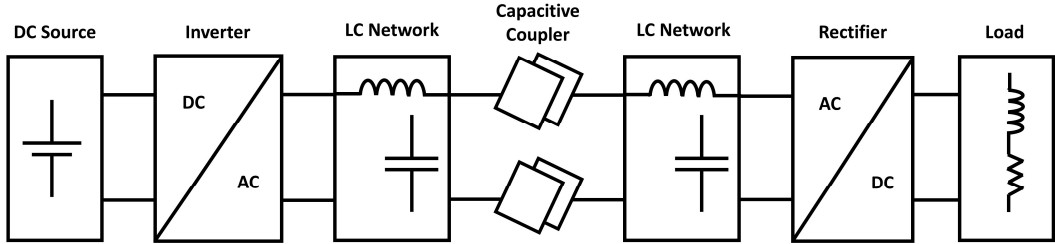

**Figure 11.** Architectural diagram of a capacitive WPT system [16].

Several topologies of CPT have been discussed in [16], where it was shown that several kilowatts (i.e., around 2.4 kW) of power could be achieved at an air gap of 150 mm. Although the CPT system provides lightweight, cost-effective couplers and less sensitive power transfer due to misalignment compared to magnetic field-coupled wireless power transfer systems, the power transfer capability of CPT is minimal due to the minimal capacitance between the road and conductive vehicle plates. The latter requires a very high frequency for effective power transfer [54,125]. This issue can be solved by reducing the impedance of the power flow channel with a high operating frequency [16].

Inductive WPT System

Inductive wireless charging utilizes Lenz's and Faraday's laws of magnetic induction to transmit power from one medium to another. Inductive charging can work efficiently for low low-power running applications [126]. Therefore, near-field magnetic field-coupled WPT (IPT) technology is the most widely employed and studied wireless power transfer system. The block diagram of IPT is depicted in the following Figure 12. This WPT is considered to be a loosely coupled transformer because its transmitting and receiving coils remain separated by a significant distance [127]. In magnetic coupled WPT, the primary transmitting coils generate the magnetic field. The receiving coils receive the generated field, then the power is transferred to the load through AC–DC rectification [16].

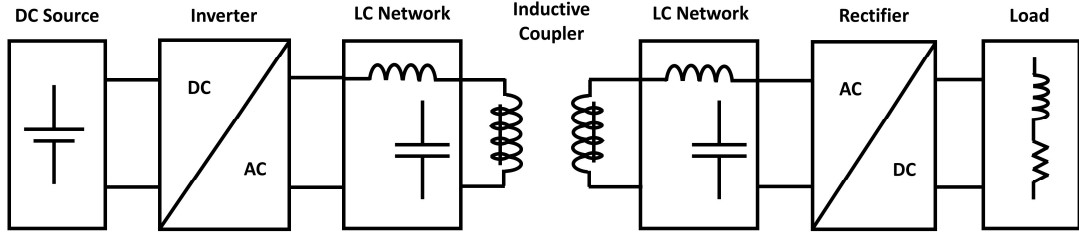

**Figure 12.** Architectural diagram of an inductive WPT system [16].

In [16], these three WPTs were compared by considering controllability, efficiency, cost, and safety, which depicted the magnetic field coupled as the most prevailing system for wireless power charging. However, the inductive WPT systems require ferrite cores for magnetic flux guidance and shielding, which makes this WPT system bulkier and more expensive [54]. Figure 13 depicts both the inductive and capacitive wireless charging technologies.

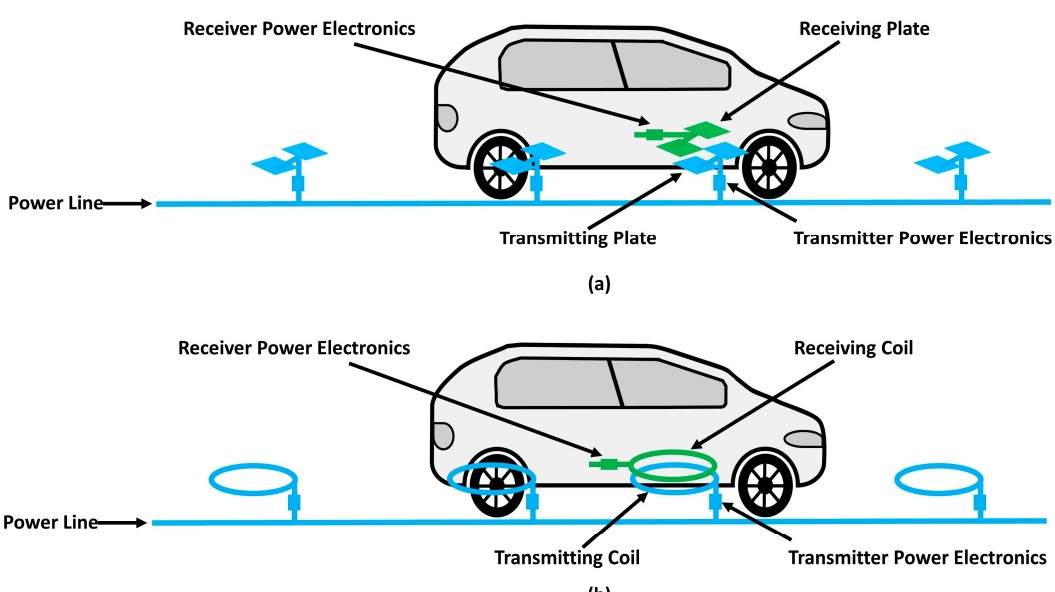

**Figure 13.** The physical implementation of two approaches to deliver energy wirelessly to electric vehicles from an electrified roadway: (**a**) inductive wireless power transfer (WPT) using coils (embedded in the roadway and the vehicle) that are coupled through magnetic fields; and (**b**) capacitive WPT using plates coupled through electric fields [54].

The near field, such as capacitive and inductive WPT, can be further divided into three sub-categories, (i) static wireless charging (SWC) [122]; (ii) quasi-dynamic wireless charging (QWC) [128,129]; and (iii) dynamic wireless charging (DWC) [130–132], which is depicted in Figure 9.

Static Wireless Charging

Static wireless charging charges the vehicle when the vehicle is stationary such as in-home garages, parking lots, etc. [122]. Static wireless charging applications were reviewed in [122], where the beginning of maturity for SWC was also concluded. One of the best advantages of using static wireless charging (SWC) is that it can be installed in convenient locations such as home garages or parking lots, eliminating the shock risk [20]. Since the alignment of SWC is enhanced, power transfer efficiency becomes more effective [7].

Quasi-Dynamic Wireless Charging

QWC operates between static wireless charging and dynamic wireless charging. When the vehicles stop for a traffic light, bus stop, or taxi stand, a quasi-dynamic wireless charging system starts charging the car wirelessly via underground fitted technology [129]. EVs can be charged with quasi-dynamic wireless charging (QWC) systems while they are stopped for a short time, such as at traffic lights. This can potentially decrease the requirements of enormous energy storage and increase EVs' overall driving range [133].

Dynamic Wireless Charging

Dynamic wireless charging charges the vehicle in motion [131]. DWC can mitigate most of the demerits of the electric car, such as small driving range, battery size, and battery cost, because there is no need to stop or wait for charging [7,132]. The dynamic wireless charging (DWC) system is famous due to its continuous charging capability. Electric vehicles are given a specified charging lane with a dynamic wireless charging system that charges the vehicles while en-routing through the lane. As a result, it increases the driving range of electric vehicle powertrains while decreasing the battery size [22]. Moreover, a conceptual demonstration of the installation of SWC, DWC, and QWC is illustrated in Figure 14 [7].

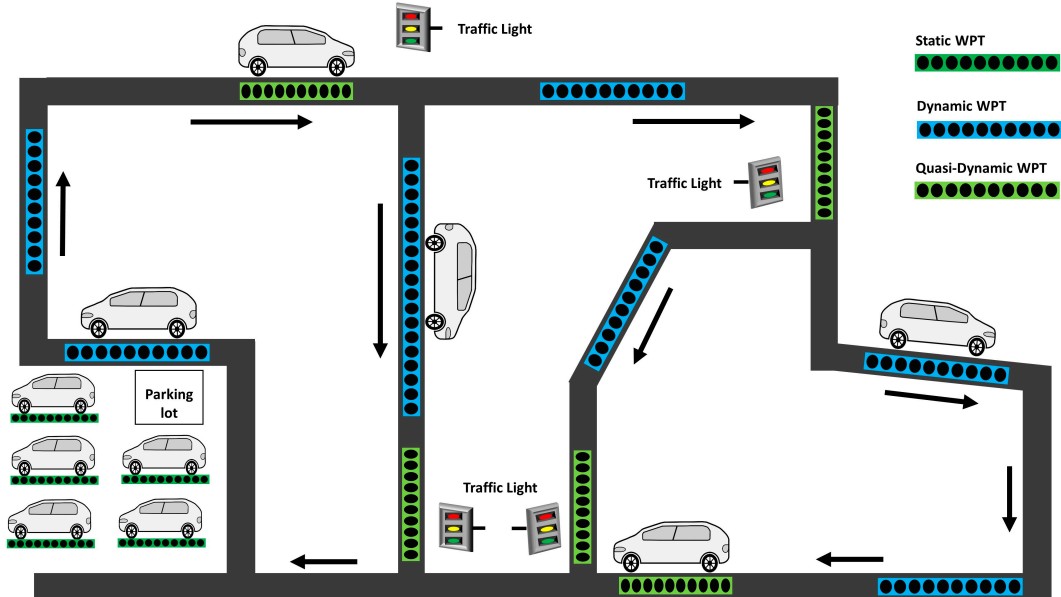

**Figure 14.** Conceptual demonstration of installation of SWC, DWC, and QWC [7].

### 2.2.2. Compensation Networks

As discussed, IPT and CPT can be considered loosely coupled transformers and capacitors due to the distance between transmitting and receiving coils and plates. Hence, due to this air gap, only a portion of the magnetic/electric flux from transmitting coils/plates is coupled to the receiving coils/plates, which causes a loss in power transfer [16,42]. As a result, passive components and switching devices face higher electric stress due to the high leakage and high circulating reactive power caused by a large air gap [130]. Consequently, a compensation network consisting of one or more passive elements is required to prevent this circulation of reactive power through a power source by providing a local path [134].

Capacitive Power Transfer Compensations

Figure 15 depicts the several types of capacitor power transfer (CPT) compensations, such as LC, LCL, LCLC, and LCL, with vertically stacked coupling plates.

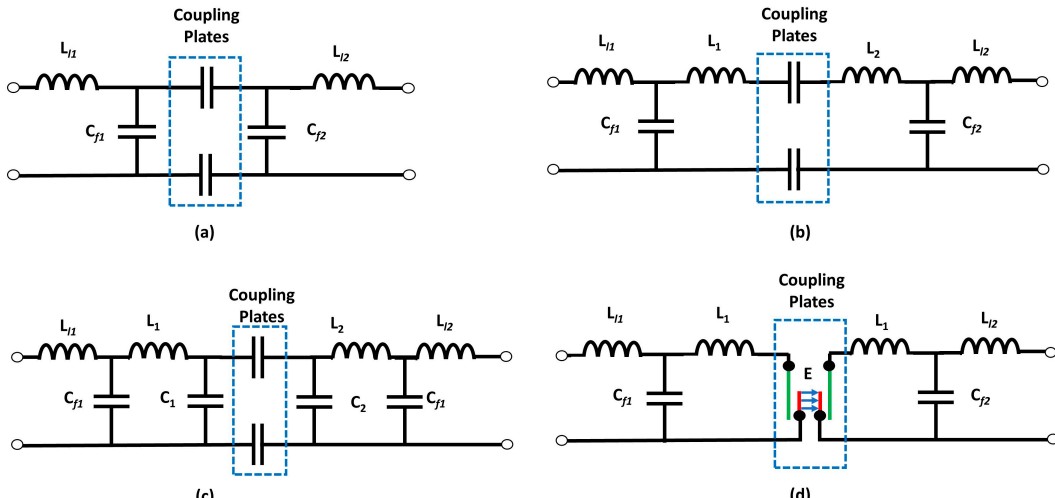

**Figure 15.** CPT compensation structures: (**a**) LC compensation; (**b**) LCL compensation; (**c**) LCLC compensation; and (**d**) LCL compensation with vertically stacked coupling plates [16].

The system efficiency for high-power operations cannot improve by LC and LCL compensations because the coupling between plates is inversely proportional to the system power. Hence, double-sided LCLC compensation is considered a suitable candidate to resolve this issue [135]. Moreover, a comparison among LCL, LCLC, and CLLC compensation capacitive wireless power transfer systems were analyzed and discussed in [136]. The LCLC compensation CPT is the best-suited topology for electric vehicles wireless charging due to its several excellent advantages, such as unity power factor, high power, high efficiency, etc. A typical structure of a capacitive wireless power transfer system with LCLC compensation is depicted in Figure 16.

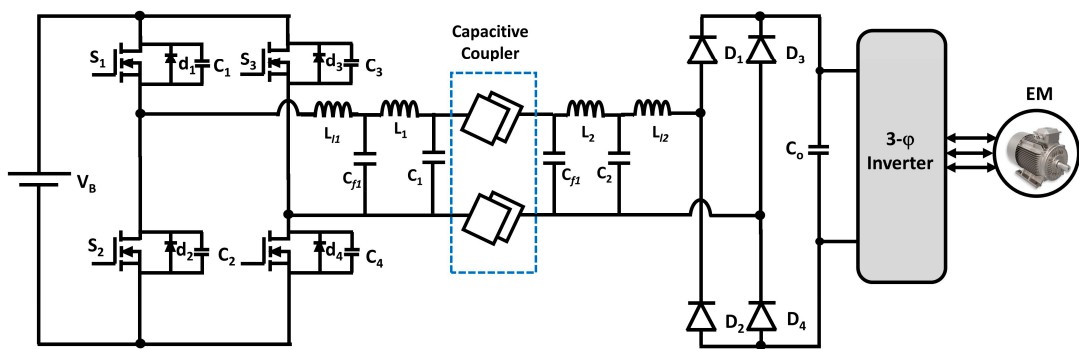

**Figure 16.** Capacitive WPT system with LCLC compensation topology [136].

In [16], CPT compensations were discussed with some detailed examples, which showed that this compensation method mainly focuses on reducing hardware complexity and enhancing power transfer capability. However, there is no unified conclusion about which CPT compensation is best suited for high-power systems due to the deficiency of large-scale practices in the CPT Field.

The design of the WPT control scheme is more complex than conventional conductive charging systems due to the trade-off between transmission efficiency and the coil/plate size of the wireless power transfer system [137]. Therefore, to achieve bidirectional power flow, degrees of freedom in terms of design, and a good performance against misalignment for an LCLC compensation capacitive wireless power transfer topology, a closed-loop control based on a phase shift control scheme was presented in [135]. The authors validated the presented control scheme via simulation results and analysis.

Some effective control schemes are also available for controlling the capacitive WPT system, including an adaptive multi-loop control scheme [138]; and a decoupled-dual-loop strategy-based control scheme [139]. The authors validated the effectiveness and efficiency of the adaptive multi-loop control scheme for capacitive wireless power charging systems via simulation and experimental results [138]. The decoupled dual-loop strategy-based control scheme can enable output power recovery under misalignment conditions and dynamic reactive compensation. Moreover, high-frequency sensing is not required to compensate for the coupling variation via a dual-loop control scheme [139].

Inductive Power Transfer Compensations

Inductive power transfer (IPT) compensation technologies are comprised of two types: (i) single-element compensations; and (ii) multi-element compensations [16].

Conventional single-element compensation technology can be categorized into four schemes, such as Series–Series (SS), Series–Parallel (SP), Parallel–Series (PS), and Parallel–Parallel (PP) [16]. Single-element compensation schemes are depicted in Figure 17.

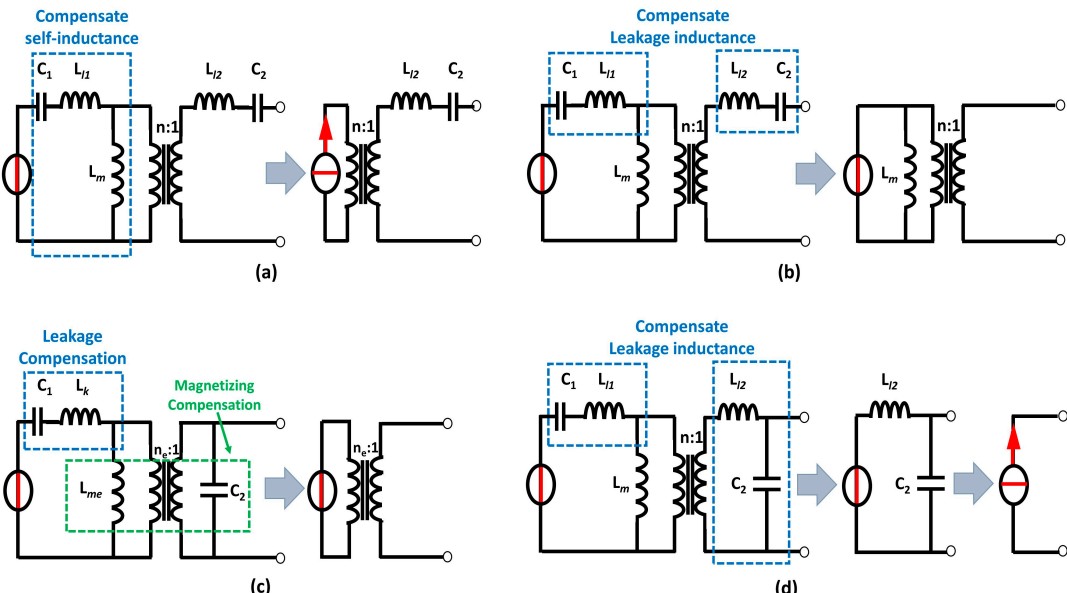

**Figure 17.** Single-element IPT compensation schemes: (**a**) self-inductance-compensated SS; (**b**) leakage inductance-compensated SS; (**c**) hybrid-compensated SP, and (**d**) leakage inductance-compensated SP [16].

Furthermore, different compensation techniques can be utilized for both primary and secondary sides [140]. In [16], only series compensation was used on the primary side or transmitting side, because a parallel capacitor cannot be used as compensation on the transmitting side due to the utilization of voltage sources for driving high-power converters. A typical structure of an inductive wireless power transfer system with SS compensation is depicted in Figure 18, which is similar to a full-bridge inverter-rectifier circuit, where $L_{11}$ and $L_{12}$ are the leakage-inductances and M is the mutual inductance between the transmitting and receiving coils of the loosely coupled transformer [141].

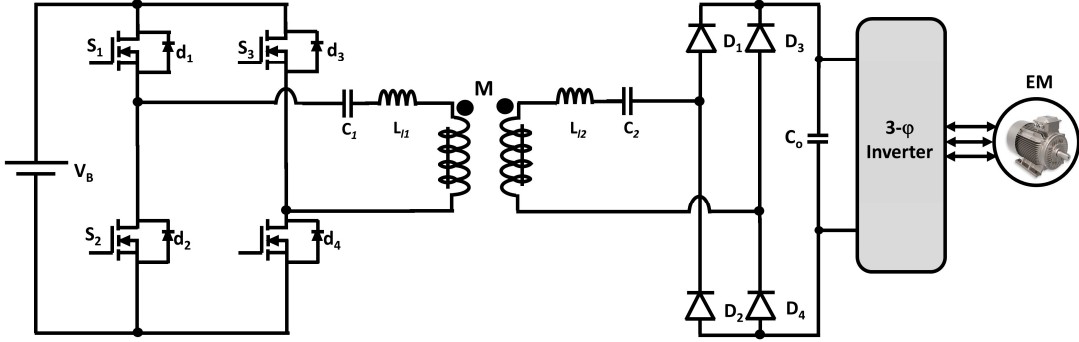

**Figure 18.** Inductive WPT system with SS compensation topology [141].

Moreover, conventional SS compensation is the most adopted high-power wireless charging system due to its simplicity, efficiency, and stability [16]. However, suppose the voltage or current source on the primary side is kept constant. In that case, the SS compensation shows a current or voltage-source behavior, depending upon the source [142,143]. Hence, to prevent the caused over-currents due to low values operation of the secondary current, an additional control scheme should be utilized to regulate the current through the primary winding [144].

In order to predict the dynamic behavior and improve tolerance to parameter variations and misalignments of the IPT system, a well-designed closed-loop controller is consistently required [145]. There are several excellent performance control schemes, such as asymmetrical clamped mode (ACM) control [145]; asymmetrical voltage cancellation (AVC) control [146]; and asymmetrical duty cycle (ADC) control [147]. Phase-locked loop (PLL) control [148] has been proposed and designed by the researchers to control series–series compensation inductive power transfer topologies. In [145], a comparative analysis among ACM, AVC, and ADC control schemes was made, where they analyzed and showed that the ACM control scheme has the least switching loss and switching frequency compared to AVC and ADC control schemes.

Several multiple-element compensations, such as T-type LCL and LCC can be utilized to mitigate the design issues faced by the single-element compensations, illustrated in Figure 19 [149–151].

Between them, the LCL compensation scheme is used more often in some applications due to its completely decoupled capability of output voltage from the load. A typical structure of an inductive wireless power transfer system with LCL multiple-element compensation is depicted in Figure 20.

Multiple-element compensations have great merits, such as minimal reactive power decoupled from load and coupling conditions and load-independent constant current/voltage output characteristics. However, due to its having more electronic components, the multiple-element compensation's complexity, cost, and size increases, and the output remains sensitive to the coupling factor [16,137]. Therefore, series–series single-element compensation is the best inductive power transfer topology between single and multi-element compensation IPT topologies [137].

The design of control schemes for multiple-element compensation IPT topologies is more complex than for single-element IPT compensation topologies due to the circuit complexity of the IPT system. Therefore, special care must be taken to design a control scheme for multi-element topologies. Following the discussed problem, a new design strategy with symmetric voltage cancellation (SVC) control scheme has been proposed and investigated in [152], to control LCL multiple-element compensation IPT topology. The authors have shown that the proposed scheme can reduce harmonics and overall cost along with an efficient switching scheme. Hence, the LCL compensation topology can achieve greater harmonics filtering capability, higher efficiency, and more constant current source properties by utilizing the proposed control scheme. In [153], to control LCC multiple-element IPT topology, a PWM feedback loop-based new control scheme was

proposed, analyzed, and showed that the control scheme can withstand high power levels and can achieve zero voltage switching for the converter.

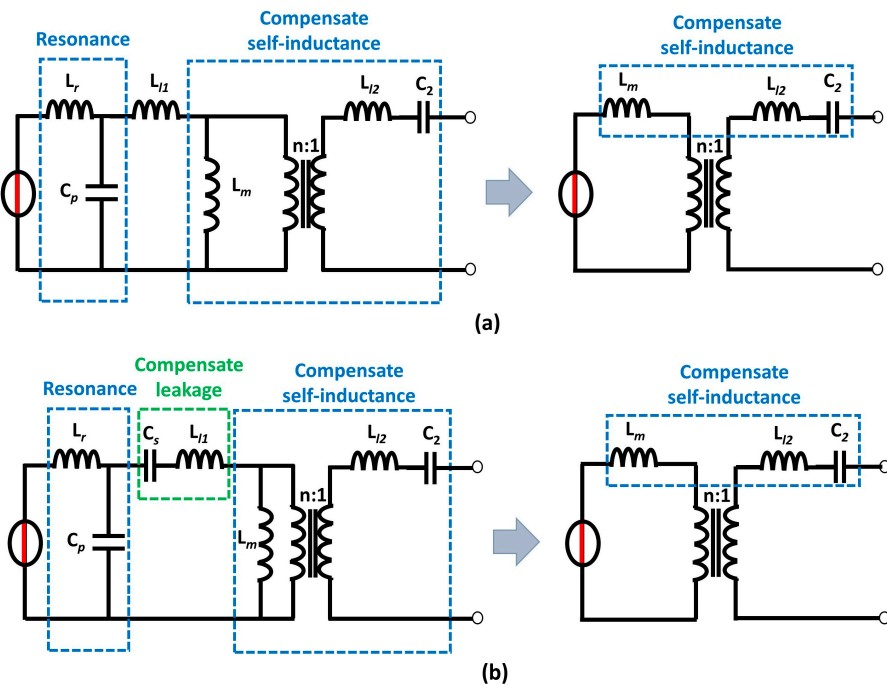

**Figure 19.** Multi-element T-type compensation scheme: (**a**) LCL (constant voltage source); and (**b**) LCC (constant voltage source) [16].

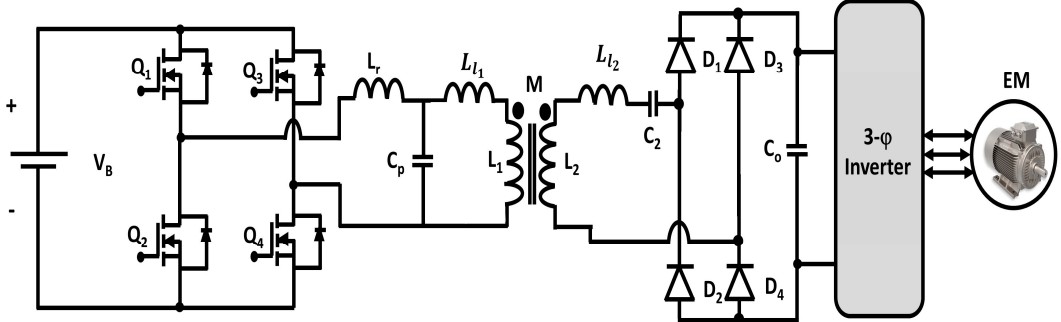

**Figure 20.** Inductive WPT system with LCL compensation topology.

### 2.2.3. Wireless Charging Challenges

The dynamic wireless charging technology of EVs is still in its embryonic stage. To implement the vision of wirelessly powered electric vehicles, numerous challenges related to safety, cost and performance must be overcome. Dynamic wireless charging of EVs has two serious challenges: (i) at high efficiencies achieving high-power transfer density while meeting electromagnetic safety requirements; and (ii) maintaining effective power transfer even as the couplers' relative position changes [54]. The power transfer density can be increased, and the size of the WPT system couplers can be reduced by designing the systems to operate at higher frequencies. On the other hand, effective power transfer can be achieved if the wireless power transfer systems are used close to the resonant frequency of the resonant tank formed by the coupler and the compensating network reactance [54].

Although wireless charging for EVs has many benefits, there are many safety issues, such as overtemperature issues due to overheating, electrical shock due to high electrical power, the intrusion of metal objects, high magnetic field exposure, and potential fire hazards [154]. In [155], symmetrical coil sets based on a foreign object detection (FOD) method were proposed to protect wireless power transfer (WPT) systems from overheating,

which could lead to an accidental fire. This type of FOD method can minimize design complexity and make the detection strategy simple to implement. In [155], a metal object intrusion detection method in the WPT system was also presented. In order to detect the presence and location of metal objects, two-layer symmetrical detection coil sets were proposed and experimentally verified, and the original magnetic field generation by the transmitter coil was taken into account when designing the symmetrical structure. The detection coils were kept in a rectangular geometry to make them easier to implement in practice. Metal object intrusion can easily be detected by comparing the induced voltage differences or mutual inductance differences between two symmetrical coils [155].

Due to the fear of electromagnetic fields, one question always arises during the WPT's deployment: "is it safe for human and animal health" [7]. In [156], it was shown that wire-less power transfer is much safer than mobile phone radiation. Furthermore, to reduce the extra exposure to electromagnetic radiation, researchers are conducting work to develop a shield for electromagnetic field shielding [7].

In [157] a secondary control method for a double-sided LCC-tuned wireless power transfer system (WPT) with an active rectifier was proposed, which improved the system performance, converter gain, and efficiency.

## 3. DC–DC Converter

For driving electric vehicles (EVs), a particular voltage level is required; otherwise, the device can be destroyed if the power is more significant than its required operating power or the device won't be able to run if the power level is deficient. A DC–DC converter is utilized to mitigate the limitation [6].

Generally, the voltage level of the battery storage and supercapacitor (SC) in electric vehicle (EV) topologies are around 250–360 V and 150–400 V, respectively, and the required operating voltage of an electric motor is about 400–750 V, which is much higher than the voltage levels of batteries and SCs. Hence, a high step-up voltage DC–DC converter is required for EV powertrains to increase the voltage level of the battery and SC. Classification of DC–DC converter topologies is depicted in Figure 21, where light-bluer highlighted topologies are well-suited for EV powertrains due to their performance characteristics [6]. In [35], comparisons between different DC–DC converter topologies have been investigated and reviewed regarding voltage-boosting techniques, applications, and efficiency.

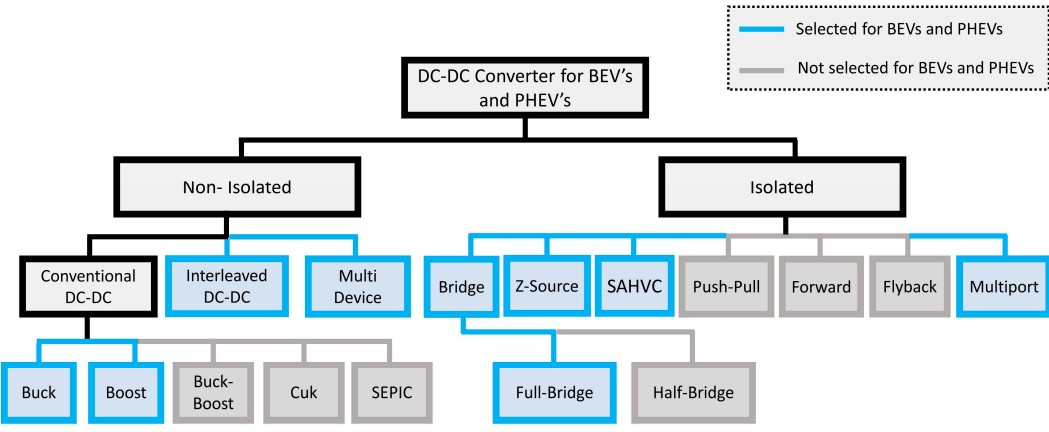

**Figure 21.** Classification of DC–DC converter topologies [6].

### 3.1. Conventional Boost DC–DC Converter (BC)

A conventional step-up or pulse-width modulation (PWM) boost converter is depicted in Figure 22, which consists of a DC input voltage source Vs., energy storage element (i.e., inductor and capacitor), controlled switch (MOSFET, IGBT, etc.) Q, diode D, filter capacitor C, and load (electric motor). In a boost DC–DC converter, the output voltage is always more significant than the input voltage, hence the name "Boost" [158–160].

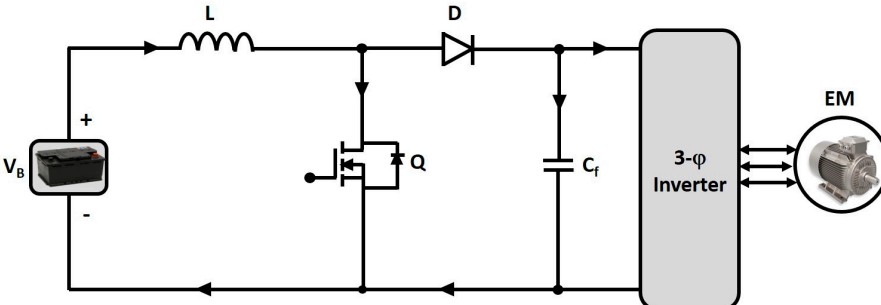

**Figure 22.** Conventional Boost DC–DC Converter.

The conventional boost converter has several merits, including simpler circuitry, lower cost due to fewer component counts, filtering and reducing electromagnetic interference efficiently, and high efficiency [35,161,162]. Despite many merits, this converter cannot achieve high voltage gain; extra protection requires to protect the circuit from short-circuit, and the power-switching devices require a parallel arrangement for handling high power, and the system is quite large in volume and heavily weighted because the large capacitor that is used to filter out the output voltage ripples [160].

To maintain a constant output voltage despite changes in input supply, designing a high-performance control system for DC–DC converters is very difficult due to the nonlinearity such as bifurcation, multiple equilibrium points, periodic behavior, and chaos of the converter [163–167]. Controllers can be of two types: voltage mode controllers, and current mode controllers. Current mode controllers are widely utilized for DC–DC converters due to their several benefits [168–171].

In [159], several state-of-the-art control system design methods, such as sliding mode control (SMC) [172,173], model predictive control (MPC) [174–176], intelligent fuzzy logic/control system [177–179], fractional-order proportional-integral-derivative (FOPID) control systems [180–182] were proposed for conventional DC–DC boost converter topologies to mitigate these problems. The SMC method has invariance to internal parameter variations, insensitiveness to external disturbances, fast transient response, and can improve the robustness quickly against nonlinear uncertainties; the MPC can easily consider the state variables/input constraints in the design procedure and control the conventional DC–DC converter. The fuzzy logic-type PID control methods are generally utilized due to their effective, simple, practical, and easily tuned capabilities [163,179]. In [163], a fractional-order PID (FOPID) control method was proposed and verified via experimental studies, which showed that the control system can provide a faster recovery time and less overshoot for a DC–DC boost converter.

### 3.2. Interleaved Four-Phase Boost DC–DC Converter (IBC)

The interleaved boost converter comprises a parallel connection of boost or step-up converters, as depicted in Figure 23. The current is divided due to a parallel connection. So, current stresses are decreased as the power losses are minimized [183]. The interleaved four-phase boost DC–DC converter (IBC) comprises four similar inductors (L1, L2, L3, L4) in four step-up levels to reduce the conductor weight and input current ripples, four parallel power-switching devices for successive phase shifting, diodes, and a filtering capacitor to eliminate the output voltage ripples. All these inductors contain individual magnetic cores for better energy storing and release. As a result, the IBC topology can increase the voltage level by more than four times [161].

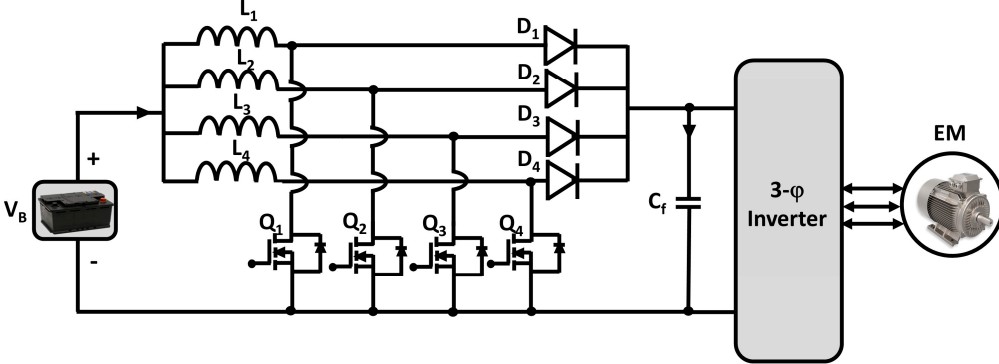

**Figure 23.** Four-phase Interleaved DC–DC Boost converter.

Moreover, the four-phase interleaved boost DC–DC converter has been chosen for EVs for its several compatible reasons, such as a reduction in inductor size and output capacitor size, a more significant reduction in input current and output voltage ripples, and higher overall system efficiency [6,13,183]. Nevertheless, the IBC is sensitive to any changes in the duty cycle, high cost, and impact on the magnetic core due to any changes in load [184].

The interleaved multi-phase boost converter has an enormous controlling process, but to make the system stable for significant disturbances [177,185–193], some advanced control methods such as model predictive control [186,187], fuzzy controller [177], sliding mode control (SMC) and PI hybrid controller [188–190], high order sliding mode control (HOSMC) [185], active disturbance rejection control (ADRC) [185,191–193] were proposed, designed, analyzed and applied to the multi-phase IBC. Moreover, an advanced hybrid Super-Twisting (ST) ADRC dual-loop controller was proposed and developed in [185], where they discussed that the ST-ADRC controller provides stronger robustness against the input voltage and load disturbances, better voltage tracking performance, and a more significant reduction in both the recovery time and voltage fluctuations compared to the conventional control systems and can improve the control performance of the converter tremendously.

### 3.3. Boost DC–DC Converter with Resonant Circuit (BCRC)

Conventional boost DC–DC converters are typically operated with hard switching, which increases the switching loss of converters in BEV and PHEV powertrains. The boost DC–DC converter topologies uses a soft-switching configuration to suppress this loss. In the soft switching technique, during the switching transition (i.e., turn ON or turn OFF), voltage or current across the switch becomes zero. As a result, the product of the voltage and current is zero; hence power losses are zero. Thus, the converter can achieve high switching frequency by reducing switching losses. Due to the switching loss reduction, the heatsink size becomes lessened, which decreases the converter volume [184]. The soft-switching configuration for the boost DC–DC converter with the resonant circuit is shown in Figure 24, which consists of two switching devices, the main switch Q1 and the auxiliary switch Q2 [6].

Moreover, any abnormality in load power will not affect the converter as the converter has high safety regulations. However, BCRC is incompatible with high-power EV powertrains and does not support bidirectionality [160,194].

Soft-switching techniques are considered the best-suited way to enhance the efficiency and reliability by reducing switching losses of electric vehicles' DC–DC converters [195]. Nevertheless, due to the necessity to exact control of numerous switches and load-dependent timing, the design of a control system for soft-switching DC–DC converters is considered complex. To meet transient requirements of voltage matching, power transfer, and response time, against system uncertainties, a robust control method is required for the electric vehicle's soft-switching DC–DC converter because the converter must work under nonlinear transient load variations in the electric vehicle [196]. In [197], a

proportional-integral (PI) controller for a soft-switching boost DC–DC converter with an auxiliary resonant circuit was analyzed and utilized, where they verified through simulation and experiment that the controller could improve the efficiency of the system. For electric vehicle soft-switching bidirectional DC–DC converter, a comparison with different time domains between PI and fuzzy logic controllers has been analyzed in [198], where they presented during settling and peak overshoot rise, fuzzy controller has better performance than PI controller.

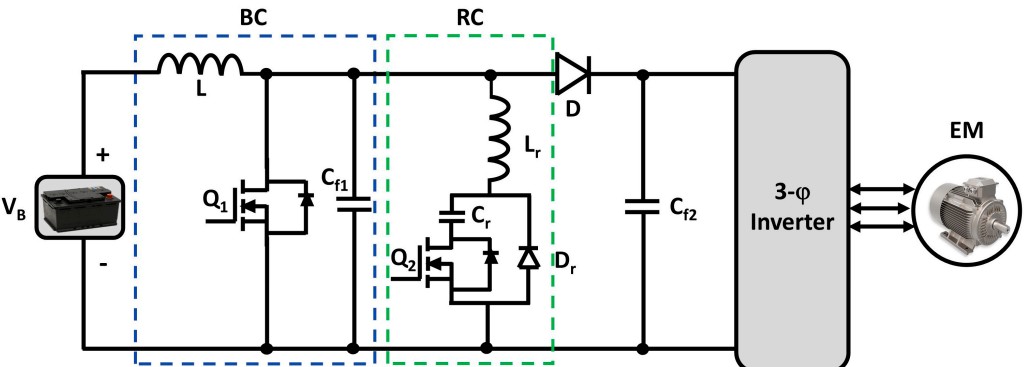

**Figure 24.** DC–DC Boost Converter with auxiliary Resonant Circuit.

Although there are several established analog controllers available in the market to control soft-switching resonant circuit DC–DC boost converters, due to the low price-to-performance ratio, high-frequency conversion system, and latest developments of micro-controllers/digital signal processors (DSP), digital controllers have been growing interest in the field of low-medium power DC–DC converter topologies [199]. However, the design of digital controllers for auxiliary resonant circuit boost converters was discussed in very few papers. In [199], some investigations were made for the digital controller design of these kinds of DC–DC converter topologies to bridge this gap. In [200], a digital control with a pole-zero placement technique was proposed, designed, and verified for a soft-switching high gain DC–DC boost converter through simulation and experimental studies. They showed that against various disturbances, the designed controller could regulate the load voltage. In [201], a robust digital PID controller for a soft-switching H-bridge boost converter is proposed and designed where they ensure step loads and source rejection with robust performance against converter parameter uncertainties, system stability, and load voltage regulation. In [199], several single-loop control methods were discussed; among them, the single-loop voltage-mode control technique is widely utilized due to its good dynamic response and simple controlling strategy. They designed the proposed digital voltage-mode controller and experimentally verified the reliability of the designed control scheme. In [196], a fixed boundary layer sliding mode control (FBLSMC) method for an electric vehicle soft-switching DC–DC converter was presented and discussed so that the FBLSMC can fix the boundary width and the instability of traditional sliding mode control can be avoided.

*3.4. Full Bridge Boost DC–DC Converter (FBC)*

The circuit diagram of an isolated full-bridge bidirectional DC–DC converter is depicted in Figure 25. This converter works in buck-and-boost mode, thus it is a bidirectional converter. In the forward direction, it works as a buck converter; in the backward direction, it operates as a boost converter [202]. Due to this bidirectionality, an isolated full-bridge bidirectional DC–DC converter can charge the batteries and provide the voltage to the load. The bidirectional full-bridge DC–DC converter has three working stages: DC–AC conversion as an inverter; step-up/step-down AC voltage with a high-frequency transformer (HFT); and AC–DC conversion as a rectifier [6].

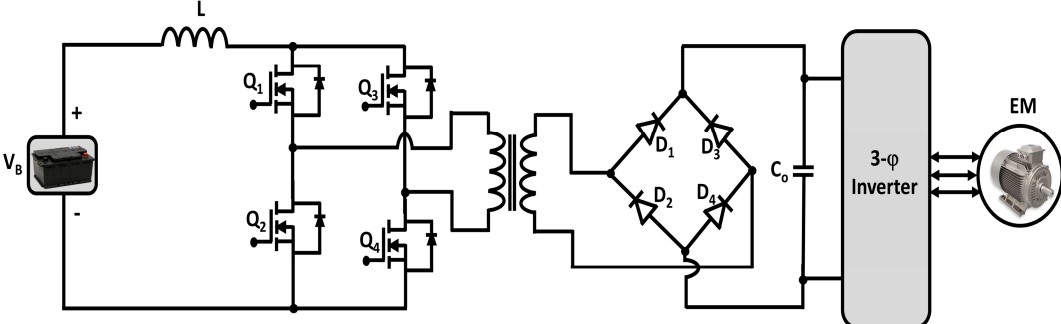

**Figure 25.** Isolated Full-bridge Boost DC–DC Converter.

Furthermore, the interior high-frequency transformer (HFT) provides galvanic isolation between the input and the output side and offers high step-up voltage. The converter can provide around 92% of efficiency at a 30 kW load [161]. However, the leakage inductance of the HFT has a crucial effect on the switching circuit due to high electrical stresses across the switching devices. Thus, the switching course required a clamping circuit to resolve the peak voltage issue [184,203,204].

As discussed earlier, digital controllers are outrunning analog controllers in the application of DC–DC converters due to several advantages, such as simplifying complicated functions and accomplishing wide-load range soft-switching. Therefore, in [205], a simple digital controller for an isolated full-bridge DC–DC converter of electric vehicle applications was presented where the driving signals for all power switching devices and the feedback control of the output power system were controlled via a single peripheral interface controller (PIC) microcomputer. For maintaining the output voltage, a digital PI control technique with a field-programmable gate array (FPGA) was presented and designed in [206]. In [186], two predictive model control (MPC), such as linear MPC (LMPC), and non-linear MPC (NMPC) techniques for an isolated DC–DC FBC, were presented and designed through both simulation and experimental results, where it was revealed that the peak current protection and the voltage regulation could be successfully achieved with both of these MPC algorithms. However, they do not assure better performance in a longer prediction horizon, and the linear MPC has a longer computational time than the non-linear MPC. In [207], a comparative performance analysis among linear peak current mode control (LPCM), non-linear carrier control (NLC), and predictive switching modulator (PSM) control schemes for isolated DC–DC FBC were presented. They proposed the PSM control scheme for IFBC due to its several advantages, such as steady-state stability performance and good transient over both LPCM and NLC schemes. Moreover, the predictive switching modulator control scheme can also reduce the rise and settling time and the peak overshoot during load changes. In addition, the PSM control scheme can reduce the size of electromagnetic interference (EMI) by extending the range of continuous conduction mode, and to generate the carrier waveform proposed controller requires only two reset integrators, whereas the NLC requires three rest integrators.

However, digital control techniques are needed to fulfill a certain condition for the resolution of the analog-to-digital converter (ADC) and resolution of PWM, otherwise, the output voltage oscillates, which is not the desired phenomenon of a controller. Furthermore, digital controllers are inherently slower than conventional analog PI controllers due to the requirements of heavy calculations. Hence, to control the isolated full-bridge DC–DC boost converter, traditional analog PI control techniques are still preferred [206].

### 3.5. Isolated ZVS DC–DC Converter (ZVSC)

For isolation, cold starting, and soft switching, an isolated ZVS DC–DC converter is needed [208,209]. Figure 26 depicts an isolated zero-voltage switching DC–DC converter (ZVSC), where a dual half-bridge topology is placed on both sides of the transformer, and for soft-switching, each power switching device has a parallel capacitor [208]. ZVSC has a sim-

ple control technique, higher efficiency, soft-switching without extra circuity, high-power density, less component count, compact packaging, and lightweight, and no real device rating consequences compared to the traditional full-bridge DC–DC converter [161,208].

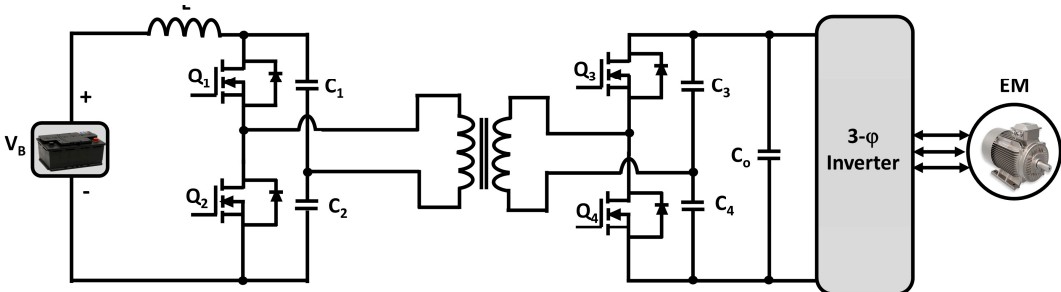

**Figure 26.** Isolated Zero Voltage Switching (ZVS) DC–DC Converter.

Nevertheless, ZVS converters are unsuitable for high-power (>10 kW) EV applications due to the absence of tolerance operation and high voltage stress remaining across the power-switching devices. The full load current must be managed across the switches ($Q_1$–$Q_4$) by dividing DC capacitors ($C_1$–$C_4$). Furthermore, for filtering the output voltage ripples, a larger capacitor is needed [208,210–213].

There are two conventional control schemes for controlling the isolated half-bridge ZVS DC–DC boost converter: symmetric and asymmetric. To control the isolated ZVSC, the conventional symmetric PWM control scheme is not a suitable candidate due to the soft-switching counterpart. For controlling the isolated ZVS boost converter, the asymmetric control scheme was proposed in [214–216]. In [214], a conventional asymmetric phase-shift control method was utilized to control the isolated half-bridge ZVSC by charging the upper and lower secondary capacitors differently. They adjusted the switching time of the secondary switches with this different charging mechanism. As a result, the voltage imbalance occurred.

Although an isolated half-bridge ZVS DC–DC converter can be controlled via a conventional asymmetric control scheme, due to the variable voltage and current equipment stresses, the asymmetric control scheme is unsuitable for wide-range input voltage [217]. As a solution in [217,218], a new asymmetric duty-cycle shifted PWM (DCS PWM) control method was proposed for an isolated half-bridge ZVS DC–DC boost converter to achieve zero voltage-switching and soft-switching behavior for all switching devices at a wide-range input voltage without adding additional components and without causing the asymmetric penalties. They experimentally verified that the proposed controller could eliminate the ringing and switching losses and operate at a higher efficiency and frequency than the conventional symmetric and asymmetric control schemes. In the DCS PWM control scheme, the lagging switch was achieved by shortening the interval between two symmetric PWM driving signals. Hence, one of the two symmetric PWM driving signals was shifted close to the other [217]. Moreover, in [219], a dual closed-loop controller of the inner balanced current loop and the outer voltage loop was designed to control and achieve the stability and robustness of a current-fed half-bridge isolated ZVS DC–DC converter of a hybrid electric vehicle. They verified the effectiveness of the designed controller with PSIM and MATLAB/Simulink simulations at various input voltages.

### 3.6. Isolated Multiport DC–DC Converter (MPC)

The isolated multiport DC–DC converter is used when more than one input source is needed with galvanic isolation between the source and load. The multiport converter is classified into three main categories: single-input multi-output (SIMO) converter, multi-input-single-output (MISO) converter, and the multi-input-multi-output (MIMO) converter. Among them, the MIMO multiport DC–DC converter is used in battery and plug-in hybrid vehicles. It has coupled multiple input sources (generally supercapacitor and battery) and

uses them as a single source with the advantages of various sources. The circuit diagram of the MIMO-MPC boost DC–DC converter is shown in Figure 27, which consists of a parallel connection of two boost DC–DC converters with bidirectional power flow, and it is advantageous for BEVs powertrains because it allows the converter to recharge the input sources during regenerative braking. Because of this, the effectiveness and functionality of the MPC increase, which makes it a high-power density converter. Moreover, all the input ports, as well as the output port, remain isolated from each other due to the interior multi-winding transformer [6,13,35].

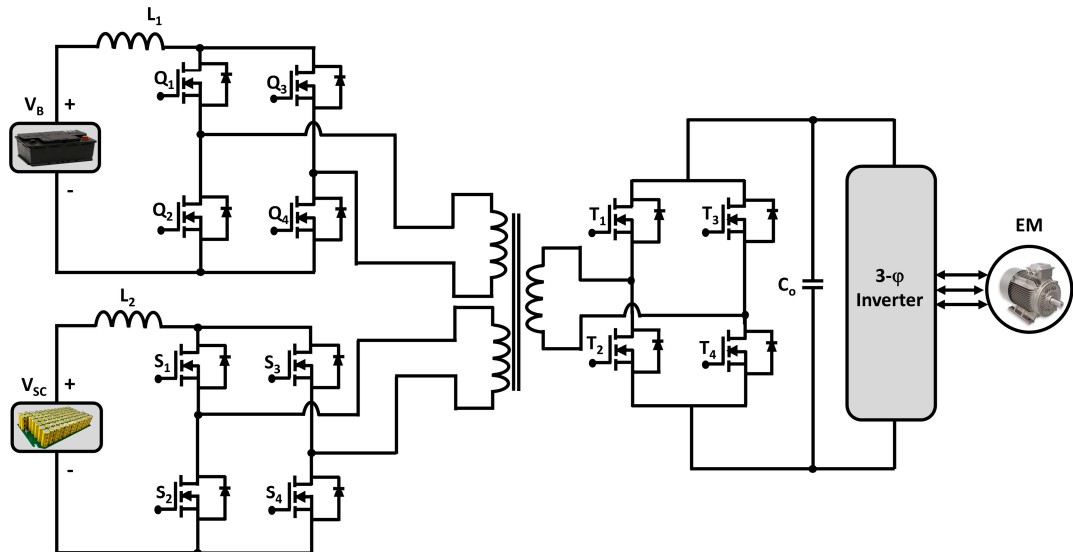

**Figure 27.** Isolated Multiport DC–DC Converter.

Nevertheless, MPC has a great component count, which makes synchronization difficult. Also, the weight of the converter increases due to the presence of the multi-winding transformer; it is sensitive to any changes in the duty cycle and analyzing the converter's steady-state and transient conditions is complex [220–223].

A control scheme is needed to combine multiple input sources, such as batteries and supercapacitors, via MPC, and is supplied to a single output. Several control schemes, such as a novel PWM plus-phase angle shift (PPAS) control scheme [224], PIC control-based scheme [225], PID control scheme [225], hybrid phase-shift and duty cycle-based control scheme [226] have been proposed and designed by many researchers to control multi-port DC–DC converters. The PAPS control scheme can achieve decoupled control and improve the device-sharing ratio among different ports [224]. Moreover, the hybrid phase-shift and duty cycle-based control scheme ensure the balance of each port power by the phase-shift control based on a reference value and the desired load voltage level kept by the duty cycle control. Therefore, this new control scheme can easily achieve the wide range of power flow control and voltage regulation [226].

### 3.7. Multidevice Interleaved DC–DC Bidirectional Converter (MDIBC)

A transformer is a heavy component that increases the weight of an isolated boost DC–DC converter. For this reason, a non-isolated multidevice interleaved bidirectional DC–DC converter is used in high-power vehicular applications. A non-isolated multi-device interleaved bidirectional DC–DC converter (MDIBC) is shown in Figure 28, which uses a battery as the primary power source and a supercapacitor as a secondary or auxiliary power source. MDIBC is a multiphase multiport bidirectional converter consisting of a phase interleaving technique with four high-frequency switching devices per phase. The number of parallel devices per phase decreases with the increasing number of phases [220].

Multidevice interleaved boost DC–DC converters merge the power from two or more input sources and are supplied at a constant single output voltage level.

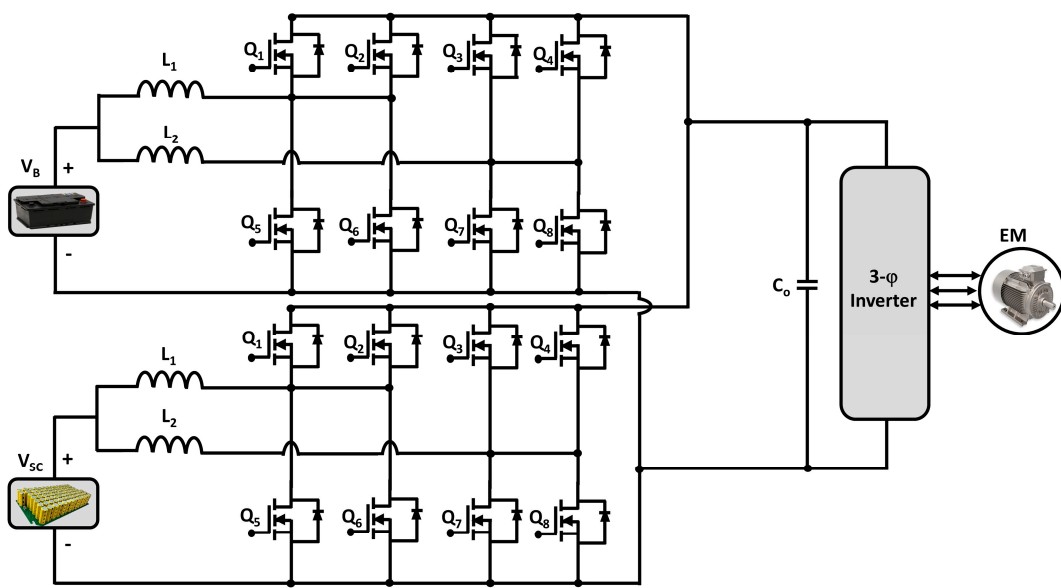

**Figure 28.** Bidirectional Multidevice Interleaved DC–DC Boost Converter.

Furthermore, MDIBC can sustain the required input current and output voltage ripples level without increasing the value of passive components (inductors and capacitors). The electrical breakdown chances are most negligible in MDIBC because a common control technique, a common heat-sink, and a standard capacitor are mutually shared by both ports, which enhances the reliability of the multidevice interleaved boost converter compared to conventional converter topologies. The overall system efficiency and effectiveness of MDIBC increases due to the regenerative braking power using the bidirectional power flow capability. Moreover, dividing the current between multiple phases is the central prominence of the multidevice interleaved boost converter. The input current ripples reduce due to the operation of the gate signals with the interleaving technique [6].

On the contrary, MDIBC has a high component count, stability, and sensitivity problem because of changes in the current load profile. Analyzing the characteristics at the transient and steady-state conditions is difficult [227–232]. Although current ripples, volume, cost-effectiveness, and the weight of vehicular power electronics interfaces are the main design challenges, bulky weight elements (filtering capacitor, inductor, and heat-sink) of MDIBC are reduced by the interleaving technique as well as by achieving high switching frequency, which fulfills the design goals [6].

For a common high-voltage DC-bus to control power regulation, the design of the controller plays an important role [228]. For controlling the multidevice interleaved DC–DC boost converter for electric vehicle applications, there are several control schemes, such as direct digital control (DDC) based digital dual-loop control [228], dynamic evolution control [233], digital phase shift control [234], advanced sliding mode control (ASMC) [235], voltage and current controllers [236], and fuzzy logic controllers (FLCs) [237,238].

To eliminate the chattering effects, which is one of the main drawbacks of the conventional sliding mode control, the ASMC controller was designed and analyzed. The elimination of chattering effects and the reduction of the voltage and current ripples with a faster transient response, stable steady-state response, and slight overshoot during startup can easily be achieved with this advanced SMC scheme [235]. Furthermore, the digital dual-loop control based on direct digital control was designed and validated via simulation and experimental results to achieve the proper regulator for the converter with a fast transient response, high performance, and high efficiency [228].

Although conventional controllers are easy and straightforward to design, due to several disadvantages, such as working point dependent performance, the stabilization problem, and control parameters need to be changed whenever the input supply and/or output parameters change, etc. Among the conventional controllers, FLCs are widely utilized in the field of power electronics MDIBC converters applications [239]. Moreover, the FLCs are simple to design and implement and are practical and powerful under parameter variations for both linear and non-linear systems [240–242]. Hence, FLCs provide better performance for traction applications than conventional controllers [238].

## 4. Energy Storage

The battery is modeled as a fixed voltage source with an internal resistance [243,244]. High performance is required for batteries since they are the core component of an EV [245]. Several types of rechargeable batteries, such as Ni–Cd, Ni–MH, Lead–acid, and Lithium–ion (Li–ion), are now available in the world markets for powering electric vehicles [40,246,247]. Among all battery types, Li–ion batteries are considered the best and are widely used in EVs due to their superior characteristics and performance, such as high energy and power density, long service life, negligible memory effect, low self-discharge rate, and environmental friendliness [247–249]. Due to these advantages mentioned above, EVs commonly utilize Li–ion batteries as the primary energy source [250–252]. Nevertheless, the biggest problems of these electrified transportations, i.e., BEVs, trucks, and buses, are the longer charging times and low driving range, averting their fast growth [17]. Hence, a higher voltage (800 V) Li–ion battery (i.e., lithium titanate, lithium nickel manganese cobalt oxide, and lithium iron phosphate oxide) is needed together with fast chargers (FCs) or extreme fast chargers (XFCs) [5,6].

In order to describe the electrical behavior of a battery, several test models exist [244]. In terms of the time required for 845 km inter-city travel, a comparison between EVs and internal combustion engine vehicles (ICEVs) was made in [4], where it was disclosed that, based on the present battery capabilities, charges with a power greater than 400 kW are required to have comparable travel time between EVs and ICEVs.

Furthermore, the advantage of XFC in EVs is that it enables a higher-voltage DC link. Higher DC-link voltage provides several benefits, such as lower manufacturing cost, higher power density, faster charging, lighter cables, lower weight, and loss for EV applications [1]. Hence, the motor current can be reduced due to the high-voltage operation which achieves high efficiency and low conduction losses [253,254]. Because of these advantages, manufacturing companies are moving toward a higher-voltage DC link. Some of the commercial electric vehicles with their battery voltages are listed in Table 2.

**Table 2.** Battery Voltage of Some EVs on the Market [1].

| Vehicle | First Production Year | Battery Voltage (V) |
|---|---|---|
| Nissan Leaf | 2010 | 350 |
| Tesla Model S | 2012 | 350 |
| Chevrolet Spark EV | 2013 | 400 |
| Audi e-tron | 2018 | 400 |
| Porsche Taycan | 2019 | 800 |
| Lucid Air | 2020 | 900 |
| Aston Martin Rapide E | 2020 | 800 |

However, despite all the merits of the high-voltage DC-link, some demerits, such as higher switching losses due to higher voltage, cannot be neglected. As a result, the overall efficiency can be decreased if the inverter topology for converting DC–AC remains the same [250]. Hence, the conventional two-level inverter is not recommended for a high-voltage DC link. Therefore, in electric vehicles, a multi-level inverter (MLI) can be a well-suited solution for high-voltage batteries [1], which will be discussed in Section 5.

## 5. Inverter

The inverter is the key component for an electric vehicle because it drives the powertrain of EVs by running the electric motor with three-phase AC from the DC-link. An inverter can be classified according to input source wise, output phase wise, output voltage wise, number of voltage levels wise, PWM wise, etc. This paper will discuss the voltage level-wise classification of the inverter. According to the number of voltages level-wise, an inverter can be classified into two main categories: (a) the two-level inverter and (b) multi-level inverter.

### 5.1. Two-Level Inverter (TLI)

The two-level inverter is depicted in Figure 29, which consists of six power-switching devices. These six power switches are connected into three legs, and each leg contains two switches, which are connected in series. An antiparallel diode is connected to each switch to allow the current to flow in the opposite direction. By controlling these six switches in different manners, the inverter can generate eight other states [243,244].

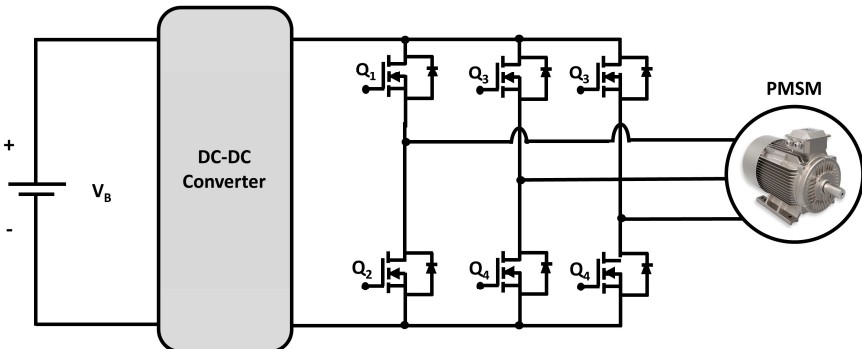

**Figure 29.** Topology of a two-level inverter (TLI).

The requirement of large filters to mitigate the output harmonics and the limited capabilities for high-power applications are the major demerits of conventional two-level inverters (TLI) [255]. Hence, to reduce the harmonics and filtering efforts, multilevel inverter (MLI) topologies were introduced [36,256].

In recent decades, power converters have become very popular for a wide range of applications, such as traction, energy conversion, and motor driver applications [257]. A new topology of a two-level voltage source inverter (VSI) was introduced in [258] that utilized a reduced number of expensive yet high-performance transistor counts, with three transistors only, having a similar performance as compared to the conventional six-transistor VSIs. Although the three-transistor VSI is designed for motor control applications, it has never been adopted for electric vehicle applications. Thus, this topology could be investigated for electric vehicle applications. Due to the high-power variable voltage and frequency supply requirement, the AC drivers are more ascendant than DC drivers [259]. Therefore, the requirement of control schemes for these power converters is also ascendant; hence, researchers present, propose, and design new schemes every year [260,261]. There are several control schemes, such as triangle comparison-based PWM (TCPWM) [258], sine-wave pulse width modulation (SPWM) [259,262], space vector-based PWM (SVPWM) [259,263], novel predictive variable structure-switching-based current controller [257], and modulated model predictive control (MMPC) [264] are available in the market of electric vehicles to control three-phase motor drive two-level power inverters. Among them, SPWM was widely utilized for its merits, such as its simple circuit, rugged and easy controllability, low power dissipation, compatibility of a digital microprocessor, and lower switching losses [262]. In [259,263], a comparison between SPWM and SVPWM control schemes for two-level inverters was conducted, showing that the SVPWM has the highest possible peak phase fundamental, less output waveform distortion, more efficient dc-bus voltage

compared to the SPWM control scheme. Therefore, space vector PWM is the best-suited control scheme to drive AC induction, brushless DC, switched reluctance, and permanent magnet synchronous motors via three-phase two-level inverters [259].

### 5.2. Multi-Level Inverter (MLI)

To gain less current and voltage total harmonic distortion (THD), less voltage stress on semiconductor devices, high efficiency, low EMI, and common-mode voltage, many conventional two-level inverters have been replaced in the past decades by their multilevel counterparts. In addition, due to the modularity and fault tolerance capability, some MLI topologies becomes more beneficial for specific applications [265,266]. The basic concept of an MLI is depicted in Figure 30, where switching devices carry much lower voltages than TLI, and the output filter size is decreased due to the ability to produce various voltage levels with better voltage/current quality at the lower switching frequency. As a result, higher power levels can be achieved with MLIs without switches derating [37].

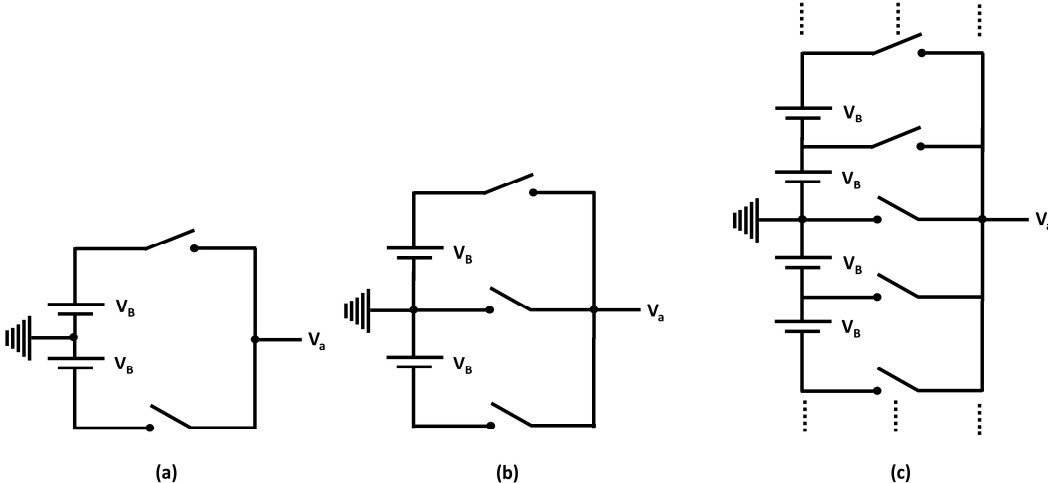

**Figure 30.** One leg of (**a**) 2-level, (**b**) 3-level and (**c**) *n*-levels inverter [37].

Although multiple-DC-source multilevel inverters (MDCS-MLIs) require multiple isolated DC supplies [267,268], in high-power motor driving applications like electric vehicles, cascaded H-bridge inverter (CHB) MDCS-MLIs are most commonly utilized due to their high modularity and identical voltage rating of the employed switches [37]. For military combat vehicles and heavy-duty trucks, cascaded H-bridge (CHB) MLI was first recommended in 1998 as a suitable choice due to it can drive high-voltage motors easily with low-voltage switching devices [269]. Furthermore, there are many different topologies, such as Neutral Point Clamped (NPC) topology [270], Packed U-Cells (PUC) topology [271], and Flying Capacitor Inverter (FCI) topology [272], by which a multi-level inverter can be built but the topology discussed in this paper is the cascaded multilevel inverter [244]. Moreover, a novel H-bridge two-transistor cascaded multi-level voltage source converter is introduced in [273], and this promising multi-level converter could be explored for electrified transportation applications.

Figure 31 depicts a multi-level inverter, which consists of series-connected H-bridges. These H-bridges can be controlled independently, and each H-bridges consists of an individual energy storage $V_{DCML}$ and four power-switching devices. The MLI can create different outputs like $V_{DCML}$, $-V_{DCML}$, 0, and open circuits by controlling these power switches in different manners [243].

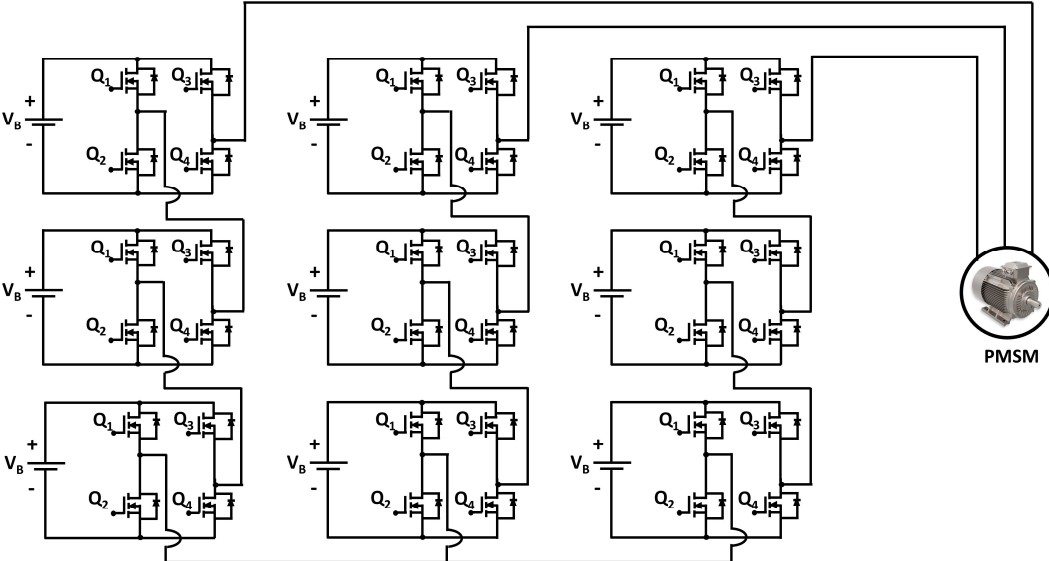

**Figure 31.** Topology of a 7-level multilevel inverter (MLI) [243,244].

Therefore, because of their remarkable characteristics, such as high-power and low electromagnetic interference (EMI), MLIs are utilized in many medium and high-power applications. However, the challenges like the suppression of circulating currents, reduction of reliability due to the higher component count (i.e., semiconductor devices and capacitors), and capacitor voltage balancing are needed to be addressed to utilize the MLI [274–276].

To control cascaded H-bridge multi-level inverter, several types of feed-forward and feed-back PWM control schemes such as sinusoidal PWM (SPWM) [277], space vector PWM (SVPWM) [277], third harmonic injection PWM (THI-PWM) [278], etc. have been presented and designed. A novel multicarrier SPWM control scheme for cascaded H-bridge seven-level inverter was proposed in [277], where the main objectives are to enhance the fundamental component of the output voltage and increase the per-phase carrier utilization by more than one. In [277], a simple SVPWM control scheme has been proposed for MLI topology where the controller utilizes a simple mapping and can be easily implemented using a microcontroller. On the other hand, the THI-PWM control scheme modifies the modulation signal and the carrier signal by adding third harmonic and phase-shifting processes to control the power converter topologies [279]. Furthermore, a performance analysis among SPWM, SVPWM, and THI-PWM in [280] where it was found that performance wise SVPWM and THI-PWM very similar, but when converter topology was considered, multi-level converters performed more efficiently with THI-PWM as compared with SPWM or SVPWM. Moreover, the DC link voltage utilization was also more than that of SVPWM.

Although the SVPWM for an MLI is complicated to implement, it is the most widely utilized control scheme for a multilevel inverter due to its several tremendous advantages such as the highest possible peak phase fundamental, more efficient dc-bus voltage, less output waveform distortion compared to SPWM, and THI-PWM [278].

## 6. Motor and Drive

### 6.1. Traction Motor

For traction applications (i.e., electric vehicles, trains, ships, aircraft), designing electric motors have hard and fast operational requirements because of high efficiency, high power density, high specific torque, low noise, fast dynamic response, high torque at low speeds, low torque at high speeds, low cost, overload capability, fault tolerance, high mechanical robustness, and ruggedness is required for traction motors [38,281]. The design should be such that the machine can produce high starting torque and low torque with high power

at high speed. Therefore, to design an electric traction motor, modeling and analysis are required from multiple engineering domains to satisfy all these requirements [253,282].

In terms of suitability for EV applications, various types of electric motors (EMs) have been presented and analyzed in [253,283–291]. Among them, permanent magnet synchronous motors are widely used due to their high efficiency and power density [292–294], depicted in Figure 32. There are several types of permanent magnet (PM) machines that can be utilized as traction motors they are surface-mounted permanent magnet synchronous machines (SM-PMSM) [295], brushless direct current machines (BLDCM) [296,297], interior permanent magnet synchronous machines (IPMSM) [298–300] for both axial and radial flux magnetic configurations [297]. However, PMSM traction motors are highly temperature sensitive. So, thermal management is a crucial aspect to design for these motors since they are expected to be performed under extreme temperature conditions as traction motors. To design and develop an accurate traction model, many parameters play an important role, such as heat transfer coefficients, boundary conditions, material properties, and geometry restrictions [301].

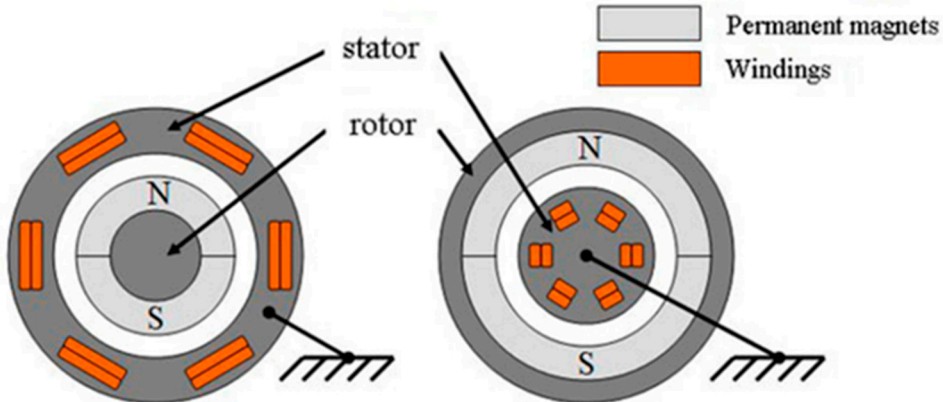

**Figure 32.** Structure of Permanent Magnet Synchronous Machine (PMSM).

Although several advantages make the PM motors a suitable nominee for EVs, the resources of the permanent magnet are limited, and the increasing demand for electric vehicles are increasing the price of the PM continuously. Moreover, the cost of the rare earth metal neodymium magnet (NdFeB) was 250 US$/Kg in 2005. On the other hand, in 2012 it was increased to 437 USD/Kg, which eventually affected the price-sensitive markets such as EVs, electric bikes, etc. [302]. Demagnetization analysis is an integral part of the design process to define the performance of a permanent magnet traction motor at a particular operating temperature [303]. Hence, the design parameters and motor geometry must be synthesized to prevent the reduction in output torque due to the irreversible demagnetization of permanent magnets [299]. From a mechanical point of view, the components of an electric motor drive system, such as bearings, couplings, and shafts, need to be analyzed to ensure reliable operation [301].

As a result, PM-free machines such as switched reluctance machines (SRMs), induction machines (IMs), and synchronous reluctance machines (SyncRels) have come into interest. Switched reluctance machine (SRM) is considered a strong candidate for electric bikes and scooters due to their robust structure and low cost [14], depicted in Figure 33.

Moreover, they can be utilized as secondary electrical power generation in electric aircraft engines due to their harsh environment operating ability [304]. In [298], several types of switched reluctance machines (SRMs) topologies such as segmental rotor SRM (SR-SRM), double stator SRM (DSSRM), and mutually coupled SRM (MCSRMs) are presented and analyzed with their performance comparison in terms of their torque density, torque ripple, power factor, and voltage utilization. In addition, the opportunities, challenges, advantages, and disadvantages of SRM drives are also discussed. In the growing electric propulsion market, because of the unpredictable cost of rare earth metals and supply chain issues of

conventional interior permanent magnet synchronous machines, the switched reluctance motor (SRM) drives have started to take their rightful place as a reliable alternative [298]. Furthermore, a new PM-less brushless synchronous machine that uses sub-harmonic magnetomotive force (MMF) to excite the rotor winding was introduced in [305,306], where the stators use novel two layers of winding in [305] and three layers in [306] for the brushless operation of the synchronous machine.

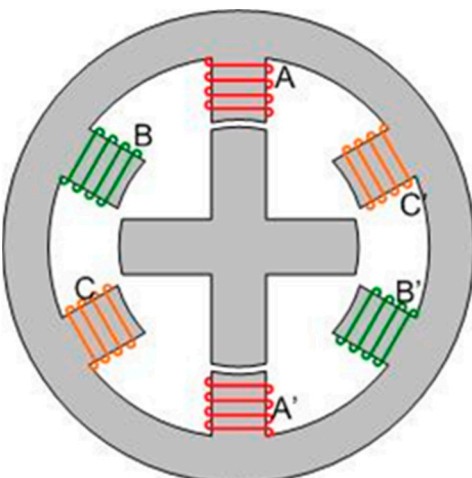

**Figure 33.** Structure of Switched Reluctance Machine (SRM).

However, although SRMs have been proposed in many research articles instead of conventional IPMSMSs due to their comparable power density/torque, there is no SRM-based powertrain for electric vehicles in the market so far [294,298,307]. Nevertheless, the overall system installation and manufacturing cost can be reduced by 30–40%, and the power density can be improved with 10–20% lesser volume by utilizing the integrated motor drive (IMD) [308,309].

*6.2. Integrated Motor Drive*

The structural integration of an electric motor drive, such as the elimination of shielded connection cables, centralized controller cabinet, and high current/voltage bus bars, is referred to as the integrated motor drive (IMD). Due to the rapidly growing interest in electric vehicles, the role of the electrified actuation system is becoming more critical. An electric vehicle requires highly efficient and reliable steering, suspension, braking, and heavy-duty actuators [310–313]. The integrated motor drive offers viable solutions for the increased demands of high-power density and highly efficient electro-hydrostatic actuator (EHA) systems [314]. In [314], four different IMD configurations were reported: (i) radially housing-mounted (RHM); (ii) axially housing-mounted (AHM); (iii) radially stator iron-mounted (RSM)l and (iv) axially stator iron-mounted (ASM), which are depicted in Figure 34a–d, respectively [314,315].

Moreover, the performance of the IMDs in high-temperature operation, fault tolerance, power density, low switching and conduction losses, lower ON-state resistance, and high efficiency can significantly be achieved by utilizing wide-bandgap semiconductor (WBGS) devices, such as gallium nitride (GaN) and silicon carbide (SiC), in motor drive technology [32,33,308,316–318]. Although the cost of WBGS-based devices is much higher than silicon-based devices due to the reduction in cooling, control cabinet, passive components, packaging, and connecting wires, the overall IMD system cost can be significantly reduced by using WBGS devices [314].

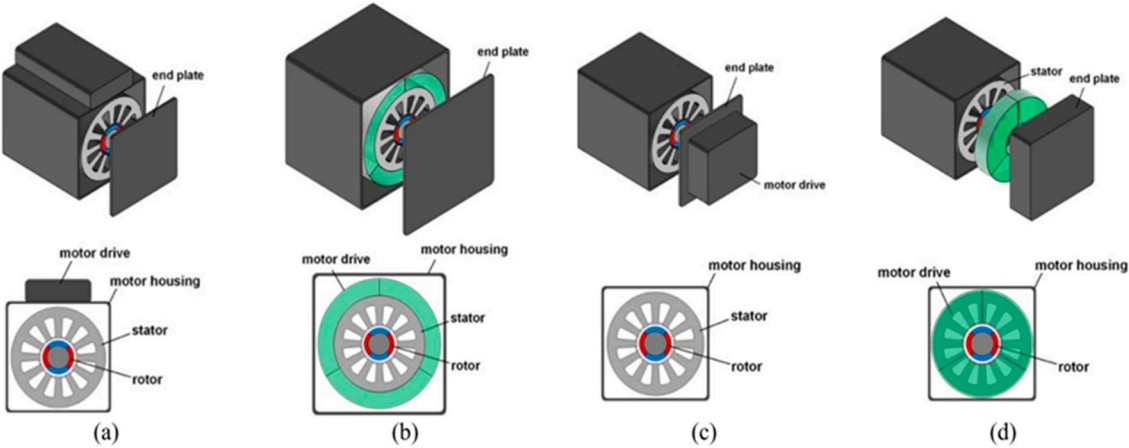

**Figure 34.** Conceptual illustration of four different IMD configurations: (**a**) RHM; (**b**) RSM; (**c**) AHM; and (**d**) ASM [15].

## 7. Simulation Result and Analysis

The simulation of all power converters, such as AC–DC, DC–DC, and DC–AC, have been conducted by utilizing MATLAB software, and the description simulation parameter values of the discussed power converters are depicted in the following tables, which are taken from both the hardware and simulation models. These design values are the most sophisticated keys for the calibration, validation and parameterization of power converters. The efficiency maps of the AC–DC, DC–DC, and DC–AC converters are observed, and the prototype test is designed based on the design values and specified parameters. The linear compensator specifications, such as cut-off frequency, bandwidth, gain, and natural frequencies, are mapped by using these design values [6].

### 7.1. Charging Section

Table 3 depicts the parameter values of input/output voltage, output power, switching frequency, inductor, capacitor, etc., for AC–DC rectifiers.

**Table 3.** Description parameter values of AC–DC rectifiers for MATLAB Simulation.

| Parameters | Bridgeless Boost Rectifier | Vienna Rectifier |
|---|---|---|
| RMS Input voltage, $V_{in}$ (V) | 230 | 230 |
| Output voltage, $V_{out}$ | 467 | 507 |
| Output power, $P_o$ (kW) | 22.41 | 32.10 |
| Frequency, f (Hz) | 50 | 50 |
| Phases, N | 3 | 3 |
| Inductor, L (µH) | - | 1000 |
| Capacitor, C (µF) | 461 | 1000 (×2) |

Although the bridgeless boost rectifier requires the least amount of components, the Vienna rectifier has a simple architecture and high efficiency. Moreover, Table 3 illustrates that, for identical supplied voltage, the Vienna rectifier provides more output voltage and power, which makes the Vienna rectifier a good candidate for fast charging in medium and high-powered EV applications. The simulation of both rectifiers was conducted with the help of a PI controller and SPWM modulation technique. The following Figure 35a,b depicts the outputcurrent–voltage waveforms of the three-phase bridgeless boost rectifier and three-phase Vienna rectifier, respectively, which also verifies the stability of the Vienna rectifier compared to the conventional bridgeless rectifier.

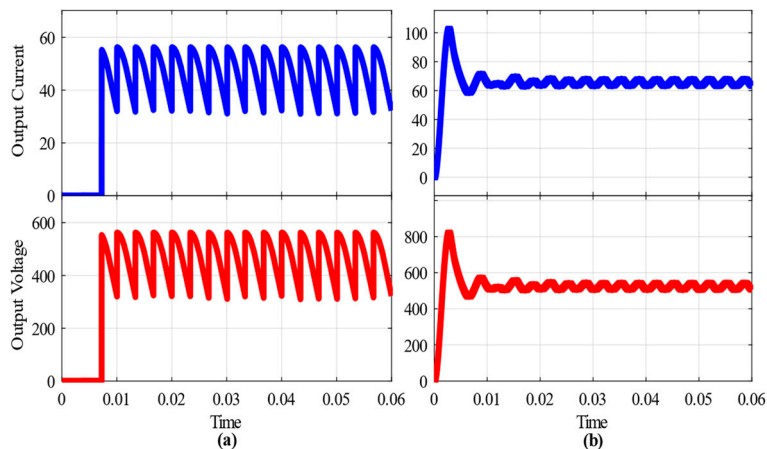

**Figure 35.** Output current-voltage waveforms of AC–DC rectifier: (**a**) bridgeless boost rectifier; and (**b**) Vienna rectifier.

*7.2. DC–DC Converter*

Table 4 depicts the description parameter values, such as input/output voltage, output power, switching frequency, inductor, capacitor, etc., for DC–DC converters.

**Table 4.** Description Parameter Values of DC–DC Converters for MATLAB Simulation [6].

| Parameters | BC | IBC | BCRC | FBC | ZVSC | MPC | MDIBC |
|---|---|---|---|---|---|---|---|
| Input Voltage, $V_{in}$ (V) | 200 | 200 | 150 | 200 | 100 | 288, 48 | 250, 200 |
| Output Voltage, $V_{out}$ (V) | 300 | 400 | 380 | 400 | 300 | 400 | 400 |
| Switching Frequency, $f_{sw}$ (kHz) | 20 | 20 | 30 | 40 | 20 | 20 | 20 |
| Inductor Current, $I_{max}$ (A) | 250 | 250 | 7.5 | 75 | - | - | 100 |
| Inductor current ripple, $\Delta I_{max}$ (A) | 12.5 | 12.5 | 0.75 | 3.75 | - | - | 10 |
| Output Voltage Ripple, $\Delta V_{out}$ (V) | 4 | 4 | 4 | 4 | 3 | - | 4 |
| Number of Phase, N | 1 | 4 | 1 | 1 | 1 | - | 4 |
| Turns ratio, n | - | - | - | 1:2 | 1:3 | 1:2 | - |
| Output Power, $P_o$ (kW) | 30 | 30 | 5 | 30 | 1.6 | 30 | 30 |
| Maximum Duty Cycle, D | 0.50 | 0.25 | 0.50 | 0.50 | 0.35 | - | 0.25 |
| Inductor, L (µH) | 400 | 100 | 6670 | 1200 | 0.56 | 175 | 187, 160 |
| Capacitor, C (µF) | 780 | 195 | 25 | 14.64 | 10 | 150 | 160 |
| Input Voltage, $V_{in}$ (V) | 200 | 200 | 150 | 200 | 100 | 288, 48 | 250, 200 |
| Output Voltage, $V_{out}$ (V) | 300 | 400 | 380 | 400 | 300 | 400 | 400 |

A qualitative analysis of all seven types of DC–DC converters for EVs and HEVs has been conducted in this literature, where the boost converter requires a simple control system, fewer switching devices, and EMI suppression. However, the BC is more significant in volume because of the use of larger capacitors; hence, it is unsuitable for high-power conversion. The output current-voltage waveforms of seven types of DC–DC boost converters have been illustrated in Figure 36, and the simulation of all these DC–DC converters has been conducted with PI controller and PWM modulation techniques.

Although IBC has a high efficiency in supplying probability at full load conditions, it has more switching losses than conventional BC. On the contrary, BCRC can reduce the switching losses due to its soft switching merits. Hence the size and weight of the BCRC can be decreased because it does not require a larger heat sink. However, like conventional BC, it cannot handle high power. Like IBC, FBC also has high efficiency (i.e., ~95%) capability at full load conditions, but the inductor volume increases due to the utilization of the high-frequency transformer. For ZVSC, switching losses are negligible because of the high frequency and power processing, and reduction in EMI suppression increases the power density. Although MPC has very high efficiency (i.e., ~98%) at full load conditions with

nominal current and voltage ripples, reliability is very low, and a robust synchronization process and EMI suppression are required.

**Figure 36.** Output current-voltage waveforms of DC–DC rectifiers: (**a**) conventional BC; (**b**) IBC; (**c**) BCRC; (**d**) FBC; (**e**) ZVSC; (**f**) MPC, and (**g**) MDIBC.

On the other hand, the MDIBC can maintain a high output voltage level without increasing the passive component's volume, which differs from the other DC–DC converters. Moreover, in MDIBC, the switches have less stress due to the interleaving technique. Therefore, the switching current rating is low, increasing the system's reliability. However, this converter's component count is high and sensitive to duty cycle changes at the load step.

*7.3. Inverter*

A 5.4 HP, 50 Hz electric motor (EM) is used as a load for the TLI, SLI, and THI-SLI topologies. It is known that the speed of EM needs to be varied in order to drive the EV properly. The variation of EM's speed is done with the help of the frequency variation, which can easily be done with the inverter. Hence, the frequency range of the TLI is from 25 Hz to 50 Hz [319]. On the other hand, the SLI and THI-SLI have the capability to vary the frequency in the range of 13–50 Hz [320]. Table 5 depicts the description parameter values such as input/output voltage, output power, switching frequency, inductor, capacitor, etc. for DC–AC inverters.

**Table 5.** Description Parameter Values of DC–AC Inverters for MATLAB Simulation.

| Parameters | TLI | SLI | THI-SLI |
|---|---|---|---|
| Input voltage, $V_{in}$ (V) | 450 | 200 | 200 |
| Output voltage ($V_{P-P}$), $V_{out}$ (V) | 450 | 400 | 400 |
| Output real power, $P_o$ (kW) | 4.6 | 4.8 | 5 |
| Frequency, f (Hz) | 25–50 | 13–50 | 13–50 |
| Phases, N | 3 | 3 | 3 |
| Output levels | 2 | 7 | 7 |
| Load, L | EM (5.4 HP) | EM (5.4 HP) | EM (5.4 HP) |

The following Figure 37a–c, depict the simulation output current-voltage waveforms of a conventional two-level inverter (TLI), seven-level inverter (SLI), and third harmonic injected seven-level-inverter (THI-SLI), respectively. The simulations of TLI, SLI, and THI-SLI have been conducted with the help of PI controller and SVPWM, SPWM, and THI-SPWM modulation techniques, respectively.

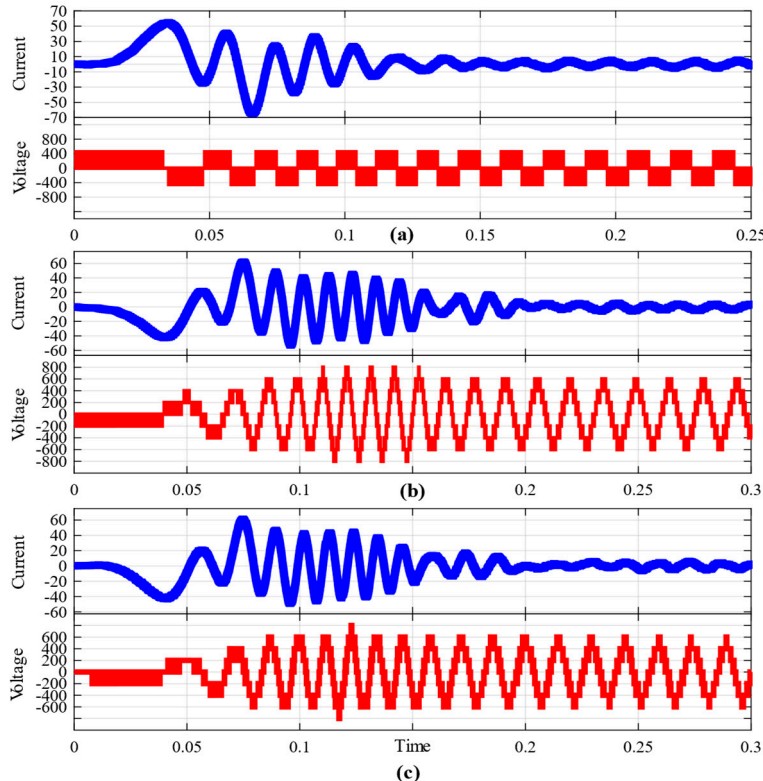

**Figure 37.** Output current-voltage waveforms of a DC–AC inverter: (**a**) two-level inverter with SVPWM; (**b**) seven-level inverter; and (**c**) seven-level THI inverter.

Moreover, MLIs are becoming more accessible and safer for electric vehicles than two-level inverters due to the benefits of higher efficiency and no electromagnetic interference [243]. It is visible from Table 5 and Figure 37 that the MLIs provide more stable and higher output power than TLI. One of the most critical advantages of MLI is that it gives much lower switching losses than the TLI because the switches of MLI change between zero or four times in the fundamental frequency period. Although a greater number of switches area greater number of switches are used in MLI, each switch is subjected to a lower voltage; hence, the losses are smaller than TLI [244].

As discussed in the charging section, the onboard chargers can be utilized in two types: FB-LLC resonant converter and PSFBC for EV charging applications. Although the full-bridge LLC resonant converter has zero reverse recovery current and ZVS, due to the requirement of a high switching frequency range to control output voltage, the efficiency of the converter becomes lower. Contrarily, the PSFBC has enormous preferable merits, such as low current stress on devices, simple control, soft-switching, and wide-range charging operations, which neglect the minor drawbacks of the PSFBC converter and makes it a suitable candidate for onboard chargers of EVs. Furthermore, between the offboard charging topologies, the Vienna rectifier seems to be a well-suited candidate for the EV charging section due to its tremendous advantages, such as fewer active switching devices, low harmonics with reliability, etc., over conventional six-switch bridgeless boost rectifiers. On the other hand, among several wireless power transfer systems, as discussed, the inductive power transfer single-element SS compensation networks provide significant advantages in high power converters over others because of the utilization of voltage

sources. Furthermore, among several DC–DC converters, the MDIBC converter appears to be the best-suited option for the DC–DC conversion process due to its higher power conversion efficiency, high reliability, high bi-directionality, and high controllability at full load conditions.

For the DC–AC inverter, the seven-level MLI appears to be best-suited because, with perfect utilization of multi-levels, the MLI can directly supply the higher voltage to the load, which can terminate the requirement of the intermediate DC–DC converter. Hence, the cost of the EV power train could be cheaper. In addition, the third harmonic injected seven-level MLI seems to be a suitable candidate for EV driving because the THI-SLI can achieve stability faster than SLI, which is also verified in Figure 37c; hence it increases the efficiency, performance, power, and voltage level of the inverter.

## 8. Comparative Analysis and Summary

A qualitative functionality comparison of two onboard charging rectifier topologies is shown in Table 6, in which five features are inspected.

**Table 6.** Functionalities comparison of onboard charging rectifiers.

| Topologies | Controllability | Bidirectionality | Reliability | Power Range | Efficiency |
|---|---|---|---|---|---|
| PSFBC | High | Not Present | High | High | Moderate |
| FB-LLC | Moderate | Not Present | Low | Moderate | Low |

A qualitative summary of the merits and demerits of onboard charging rectifiers is illustrated in Table 7.

**Table 7.** Summary of onboard rectifiers.

| Topologies | Merits | Demerits |
|---|---|---|
| PSFBC | • Simple control techniques<br>• Low current stress on devices<br>• Soft switching or zero voltage switching<br>• Low switching noise<br>• Wide-range charging operation<br>• Low EMI | • Rectifier bridge face high voltage stress<br>• High-circulating current in the freewheeling interval |
| FB-LLC | • Zero reverse recovery current<br>• Zero voltage switching<br>• High efficiency at high voltage operation | • Low Efficiency<br>• Complicated control techniques<br>• Complicated filter and transformer design<br>• High components count<br>• High switching frequency range to control the battery's output voltage |

After the inspection of both of these Tables it is visible that the PSFBC rectifier is suitable for EV onboard charger due to its high controllability, high reliability, high power range and high efficiency.

A qualitative functionality comparison and summary of merits and demerits of two most widely utilized offboard charging rectifier topologies is shown in Tables 8 and 9, respectively.

**Table 8.** Functionalities comparison of offboard rectifiers.

| Topologies | Controllability | Bidirectionality | Reliability | Power Range | Efficiency |
|---|---|---|---|---|---|
| Bridgeless Boost Rectifier | Moderate | Present | Moderate | Moderate | High |
| Vienna Rectifier | High | Not Present | High | High | High |

**Table 9.** Summary of offboard rectifiers.

| Topologies | Merits | Demerits |
|---|---|---|
| Bridgeless Boost Rectifier | • Good voltage regulation<br>• High efficiency<br>• Small input current filter, and output voltage filter size<br>• Fast switching | • Low voltage gain<br>• Not suitable for the high-power application<br>• High EMI<br>• High switching noise<br>• Pre-regulators required to control the PF |
| Vienna Rectifier | • Fewer active switching devices<br>• Capable of fast charging<br>• Low reverse recovery current losses<br>• High power rating<br>• High efficiency<br>• Does not required extra pre-regulator<br>• Low current harmonics | • High cost<br>• High components count<br>• Unidirectional |

After the inspection of the above Tables 8 and 9, it is visible that the Vienna rectifier is suitable for the EV to offboard charger due to its high controllability, high reliability, high power range and high efficiency.

A qualitative functionality comparison and summary of seven popular DC–DC converter topologies are shown in Tables 10 and 11 below.

**Table 10.** Functionalities comparison of DC–DC converters [6].

| Topologies | Controllability | Bidirectionality | Reliability | Power Range | Efficiency |
|---|---|---|---|---|---|
| BC | High | Not Present | Moderate | High | Low |
| IBC | High | Not Present | Moderate | High | Moderate |
| BCRC | High | Not Present | High | Low | Low |
| FBC | High | Not Present | High | High | Moderate |
| ZVSC | Moderate | Not Present | High | Low | High |
| MPC | Moderate | Present | Low | High | High |
| MDIBC | High | Present | High | High | High |

**Table 11.** Summary of DC–DC converters [6].

| Topologies | Merits | Demerits |
|---|---|---|
| BC | <ul><li>Simple circuit</li><li>Low cost</li><li>Simple control techniques</li></ul> | <ul><li>Large filter size</li><li>High ripple rate</li><li>Low voltage gain</li></ul> |
| IBC | <ul><li>Low input current ripples</li><li>High voltage gain</li><li>Low filter size</li><li>Simple control techniques</li></ul> | <ul><li>High switching losses</li><li>High components count</li></ul> |
| BCRC | <ul><li>Low heat sink size required</li><li>Soft switching</li><li>Low EMI</li></ul> | <ul><li>Low voltage gain</li><li>Not suitable for the high-power application</li></ul> |
| FBC | <ul><li>High voltage gain</li><li>High Efficiency</li><li>Low voltage stress on the switches</li><li>High isolation</li></ul> | <ul><li>Switching circuit faces high current stresses</li><li>Large filter size</li></ul> |
| ZVSC | <ul><li>Low switching losses</li><li>Low EMI</li><li>High power rating</li></ul> | <ul><li>Gates has a high current rating</li><li>Cannot tolerate fault</li><li>Large filter size</li></ul> |
| MPC | <ul><li>Low output voltage ripples</li><li>High voltage gain</li><li>High isolation from all input sources</li><li>Bidirectional power flow</li></ul> | <ul><li>High component count</li><li>Difficult to Synchronize</li><li>Responsive to duty cycle changes at the load step</li><li>Difficult to analysis throughout transient and steady-state conditions</li></ul> |
| MDIBC | <ul><li>High Efficiency</li><li>Switches have low current stress</li><li>The ability to deliver high power while ensuring the required output voltage level</li><li>Small filter size</li><li>Small heat sink size</li><li>Simple control techniques</li><li>Bidirectional power flow</li><li>Appropriate for single-port converters</li></ul> | <ul><li>Responsive to duty cycle changes at the load step</li><li>High component count</li><li>Difficult to analysis throughout transient and steady-state conditions</li></ul> |

After the inspection of above Tables 10 and 11 it is visible that the MDIBC converter is suitable for the EV powertrain converter due to its high controllability, bidirectionality, high reliability, high power range and high efficiency.

A qualitative functionality comparison and summary of the merits and demerits of three popular inverter topologies is presented below Tables 12 and 13, respectively.

**Table 12.** Functionalities comparison of inverters.

| Topologies | Controllability | Bidirectionality | Reliability | Power Range | Efficiency |
|---|---|---|---|---|---|
| TLI | Moderate | Present | Moderate | Low | Low |
| SLI | High | Present | Moderate | High | High |
| THI-SLI | High | Present | High | High | High |

**Table 13.** Summary of inverters.

| Topologies | Merits | Demerits |
|---|---|---|
| TLI | <ul><li>Good voltage regulation</li><li>High efficiency</li><li>Small input current filter, and output voltage filter size</li><li>Fast switching</li></ul> | <ul><li>Low efficiency</li><li>Low power rating</li><li>Cannot produce various voltage levels</li></ul> |
| SLI | <ul><li>Less voltage stress on semiconductor devices</li><li>High efficiency</li><li>Low EMI,</li><li>Common mode voltage</li><li>Modularity and fault tolerance capability</li><li>Can produce various voltage levels</li><li>High power rating</li></ul> | <ul><li>High cost</li><li>High components count</li><li>Complex circuit</li></ul> |
| THI-SLI | <ul><li>Third harmonic injection capability</li><li>Can stable faster than SLI</li><li>Less voltage stress on semiconductor devices</li><li>High efficiency</li><li>Low EMI,</li><li>Common mode voltage</li><li>Modularity and fault tolerance capability</li><li>Can produce various voltage levels</li><li>High power rating</li></ul> | <ul><li>High cost</li><li>High components count</li><li>Complex circuit</li></ul> |

After the inspection of above Tables 12 and 13 it is visible that the THI-SLI is best-suited for EV powertrain inverter due to its high controllability, bidirectionality, high reliability, high power range and high efficiency.

## 9. Future Trends

To reach the goal set by the US Department of Energy (DOE) and to increase the utilization of EVs, the challenges, such as limited charging infrastructure, range anxiety, battery pack size, and costs faced by the EVs need to be mitigated. Multiple power electronic converters (i.e., DC–DC converters, inverters, and rectifiers) are required to charge the EVs, HEVs, and PHEVs batteries from the power grid and drive the motor from the battery pack, which is depicted in Figure 1 [321–324]. Hence, these converters need higher efficiency and ruggedness with compact sizes and low costs. [9]. Moreover, to achieve the target for 2025 of 100 kW/L power converters for EVs, the Department of Energy (DOE) has identified various effective strategies. To overcome the challenges associated with designing the converters, further integration of the power converters' subcomponents and replacing silicon semiconductors with wide bandgap semiconductors (WBGSs) are considered to be necessary [325].

*9.1. WBG Devices*

In power electronic applications, silicon (Si) based devices were widely used in past years but due to their inherent properties, they have reached the limit of maximum switching frequency and device thermal dissipation [326]. Therefore, other semiconductors are required to overcome the limitation of Si. Hence, wide bandgap semiconductors (GaN and SiC) are now of interest due to their enabling ability of higher breakdown voltages and higher switching speeds, which permit higher efficiency, higher temperature operation, lower losses, and higher power density with a smaller size and weight compared to the silicon-based system [12,327–330]. Due to the characteristics and commercialization progress of silicon carbide (SiC) and gallium nitride (GaN), they are considered the most promising WBGSs nowadays [34]. In [331–334], several reviews on the applications of wide bandgap semiconductors for electric vehicles were conducted and showed that GBGS devices offer many advantages over existing silicon devices. In [32,33,335–342], several comparisons were made between wide bandgap semiconductors and silicon semiconductor-based MOSFET and IGBT, where WBGS showed tremendous advantages, such as a faster switching frequency, lower loss, and higher efficiency over silicon semiconductors.

The comparison among GaN, SiC, and Si is depicted in Table 14, where it is demonstrated that, in terms of electron mobility, breakdown of the electric field, and energy gap, GaN is higher than SiC and Si. However, GaN has much lower thermal conductivity than SiC and Si. So, heat propagates poorly from the junction to the heatsink for GaN, which makes it temperature-sensitive [343].

**Table 14.** Parameter Comparison of Si, SiC, and GaN [43].

| Parameters | Si | SiC | GaN |
|---|---|---|---|
| Electron mobility ($cm^2$/V$\times$s) | 1400 | 900 | 1800 |
| Energy gap (eV) | 1.12 | 3.26 | 3.5 |
| Breakdown electric field (MV/cm) | 0.3 | 3 | 3.3 |
| Thermal conductivity (W/cm$\times$K) | 1.5 | 4.9 | 1.3 |
| Saturation drift velocity (Mcm/s) | 10 | 27 | 27 |

GaN has lower conduction loss and smaller chip size than Si, and SiC due to its lower on-resistance, allowing GaN-based devices with simpler cooling systems and heatsinks [344–346]. However, currently available GaN devices are much higher in cost than Si. Good advancements in GaN technology and mass production can decrease the cost count in the near future [347].

### 9.1.1. Chargers

Usually, the onboard chargers consist of two converters. They are AC–DC rectifiers and DC–DC converters. Among several possible AC–DC rectifier topologies, the phase-shifted full-bridge converter (PSFBC) is commonly utilized due to its simple control, soft switching, and wide-range charging operation capability. Hence, GaN devices with a 650 V voltage rating can be used in the typical 390 V AC/DC rectifier, depending on the design requirements. Furthermore, a DC- DC converter is required to match the voltage levels of the AC–DC PFC inverter and the battery pack. The DC–DC converter topologies can achieve a high switching frequency while decreasing the overall system losses by utilizing wide bandgap semiconductor switching devices. As a result, the dynamic performance of the power converters can be improved [43].

Moreover, for fast charging offboard chargers, among a few topologies such as bridge-less boost rectifier, Vienna rectifier, and neutral point clamped (NPC) rectifier, the Vienna rectifier is generally utilized for higher adequate current ratings. GaN could also be beneficial in this fast charger [43].

Inductive chargers or wireless chargers are expected to be the future chargers for electric vehicles, which can be achieved via electromagnetism of the coils of wires. With wireless charging technology, electric cars can be charged while driving or while parked in the parking lot [344]. WBG semiconductor devices have demonstrated tremendous advantages in WPT technology over silicon-based devices [16]. A grid-to-battery efficiency of 91–93% has been achieved by building a single-phase WPT charger with SiC MOSFET and SiC diodes [28]. Moreover, several WPT chargers have been proposed and built with SiC and GaN-based semiconductor devices in [30,348–351], where the WBG overpowers Si-based devices every time in terms of high efficiency, high power, and high switching frequency.

### 9.1.2. Electrified Powertrain

The DC–DC boost converters and the DC–AC inverter are the key components for an EV powertrain, because in order to drive the electric motor with a battery, a DC–DC converter is required for stepping up the DC battery voltage. The inverter is required for inverting the DC voltage into AC [322,352].

In EV powertrains, silicon-based insulated-gate bipolar transistors (IGBTs) or metal oxide semiconductor field-effect transistors (MOSFETs) of 650 V ratings are commonly used in conventional DC–DC converters and inverters. Still, the switching frequency of these Si-based transistors is limited to around 10–20 kHz, since these converters are mainly utilizing electric motors as loads. Furthermore, the limitation of these Si-based switching devices can be eliminated by using a hybrid mix of GaN and Si, which can be beneficial with the advantages of both semiconductors [338,353,354]. Therefore, high breakdown voltage, high current, high switching speed, and high power with lower losses and costs can be achieved with hybrid GaN and Si-based transistors [43]. Moreover, the reverse recovery current and losses are eliminated by the gallium nitride high-electron-mobility transistors (HEMTs) due to the absence of a body diode. The electric vehicle's powertrain can be more compact at a lower cost because of the wide bandgap semiconductors' higher temperature handling capability [355].

### 9.1.3. Motor Drives

Both the inverter and motor are considered when looking at the motor drives because the power electronic equipment's efficiency determines the electric vehicle's efficiency. Utilizing WBG semiconductors in the motor drive application could bring high efficiency, high-temperature operation, fault tolerance, power density, lower ON-state resistance, and low switching and conduction losses with a reduction in overall cost, which will make EV a high competitor with internal combustion engine vehicles [15,43].

Although the cost of WBGS-based devices is much higher than the silicon-based devices, due to the reduction in cooling, control cabinet, passive components, packaging,

and connecting wires, the overall IMD system cost can be significantly reduced by using WBGS devices. However, manufacturing complexity, such as current collapse, reliability, and packaging, are the main constraints that stop the vast market penetration of WBGS devices [15].

*9.2. System Integration*

The structural integration of an electric vehicle's internal components is called system integration. In the production of electric vehicles, integrating power converters and electric motors has been a straightforward solution to reduce the overall system cost and increase its reliability and efficiency. In [315,356], a further increment of the traction inverter integration was proposed, in which the inverter electronics are mounted with the electric machine axially, which allows the inverter phase terminals to connect with the motor winding directly, reducing the weight, volume, and cost of the traction inverter [12]. Therefore, the integration of the EV's internal components, such as its power converters, and the electric motor, seems advantageous.

Although wide bandgap semiconductor technologies can enable may important features, such as higher switching frequency, higher efficiency, and higher power, in different applications, challenges such as complex control schemes, high cost, thermal management, and limited voltage ratings need to be improved to ensure the long-term reliability of these devices [16,43]. Moreover, dual bridge matrix converter topologies presented in [357] could be utilized in system integration. In addition to that, the power electronic converter topologies shown in [358–361] for various applications can also be implemented. Furthermore, to ensure secured power system integration, methods presented in [362–367] may also be applied for a safer and secured power system and its communication networks.

## 10. Conclusions

This review paper discussed several conductive charging rectifiers, powertrain DC–DC converters, motor-driving inverters, and control schemes designed for EV applications based on several publications. To develop more efficient and environmentally friendly EVs, power converters such as AC–DC, DC–DC, and DC–AC are the critical applications of modern power electronics. After this extensive review, it can be seen that the charging section onboard PSFBC and offboard Vienna rectifiers present great performance, and the powertrain high-power DC–DC converter section and the multi-device interleaved DC–DC boost converter seem to be excellent candidates for EV applications. On the other hand, for the powertrain DC–AC inverter, the third harmonic injected seven-level MLI can be considered a well-suited candidate for the powertrain of electric vehicles.

Moreover, the current status, opportunities, challenges, and applications of wireless power transfer in electric vehicles, and hybrid electric vehicles, were discussed in this review paper. Wireless charging is the key solution to the range anxiety and recharging times of electric vehicles. A high-power fast wireless charging system can experience fueling times similar to conventional fossil fuel refueling times. Furthermore, the wireless charging system is more reliable, does not pollute the environment, and is safer than conductive charging systems. The ever-growing application of high-power wireless chargers can achieve the mass-market penetration of EVs.

The advantages, such as faster charging, lighter cables, more efficient motors, etc., and disadvantages, such as high voltage stress on semiconductor devices and higher EMI of the traction drive system with a high-voltage battery module, were also discussed in this paper. The comparison between several conventional interior permanent magnet synchronous machines and switched reluctance motors (SRMs) was investigated in this review. It has been shown that the double stator SRM (DSSRM) drives have started to take their rightful place as a reliable alternative due to their high torque/power density, lower vibration, and acoustic emission advantages.

Finally, to increase the operating temperature profile and power density, the power converters are adopting newly introduced wideband gap semiconductors (WBGSs), con-

verter topologies, and control schemes. For automotive power electronic systems, wide bandgap semiconductor devices have been of interest for some time. With the arrival of the Tesla Model 3 with a silicon carbide (SiC) based traction inverter, WBGSs are now beginning to see the adoption. A comparison among Si, SiC, and GaN was conducted in this paper and showed that for lighter designs, without compromising the efficiency of the electrified vehicles, GaN is a well-suited candidate due to its high-power density, high breakdown voltage, and high switching frequency. GaN can make the overall system smaller and lighter in terms of its higher switching transitions compared to SiC and Si. Nevertheless, the major demerit of WBGSs is the high cost, which can be reduced by increasing the production and adoption of WBGS devices.

**Author Contributions:** Conceptualization, R.I., S.M.S.H.R. and O.A.M.; Formal analysis, R.I. and S.M.S.H.R.; Funding acquisition, R.I. and S.M.S.H.R.; Investigation, R.I. and S.M.S.H.R.; Methodology, R.I. and S.M.S.H.R.; Software, R.I. and S.M.S.H.R.; Supervision, S.M.S.H.R. and O.A.M.; Writing— original draft, R.I. and S.M.S.H.R.; Writing—review & editing R.I., S.M.S.H.R. and O.A.M. All authors have read and agreed to the published version of the manuscript.

**Funding:** This research received no external funding.

**Data Availability Statement:** Not applicable.

**Conflicts of Interest:** The authors declare no conflict of interest.

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
