# Peer review of "Comprehensive Review of Power Electronic Converters in Electric Vehicle Applications"

_forecasting, doi:10.3390/forecast5010002_

Round 1

Reviewer 1 Report

Since the whole contents consist of various topics, each topic appears poor. At least, a top framework expression including one of model based systems engineering should be  described.

Author Response

We would like to thank you for your time, valuable comments, and suggestions regarding the submitted article. We believe the changes done through the process will make the article even uplift the quality of the work. It is to be noted that all changes made in the manuscript are highlighted. 

Reviewer 2 Report

I certify that the paper could be considered for publication. Most of the manuscript is a summary of the currently used methods, for which the analysis and discussions are very brief and focused. Even though the authors included some additional input, I can attest that a survey paper for a journal requires more insightful remarks and debate.

The manuscript doesn't touch on forecasting in any way. The text is not made pertinent to forecasting and its applications by including the word "FORECASTING" in the abstract. It made things hazier. Upon reading the manuscript, I discovered that not even the surveyed papers used FORECASTING. The authors should cite the definition of FORECASTING.

The authors must present a detailed listing of the currently available literature with a summary of the method and evaluation metric using a table to assist readers’ understanding.

https://doi.org/10.1016/j.aej.2021.11.023

Author Response

(The authors gave the same response as above.)

Reviewer 3 Report

This paper aims to value the importance of power electronic converters and their control in the forecast of electric vehicle applications. This is quite a long review article, yet of good readability and technical insight. This paper is acceptable for publication if the following issues can be well addressed.

1) Do not hesitate to conduct quantitative analysis. For example, when discussing the HVDC bus, you can directly give its voltage level.

2) In addition to a detailed description of those are most commonly used, other existing methods/topologies/designs should also be mentioned "briefly". For example, except for phase-shift control, there is also frequency tracking, duty-cycle modulation, burst, etc.

3) "Off-board charging, often called fast charging." This might not be correct since fast charging can also be realized in an onboard way.

4) Concerning the EV charging schedule, the following publication might help.

-Integrating plug-in electric vehicles into power grids: A comprehensive review on power interaction mode, scheduling methodology and mathematical foundation

5) What is the frequency level of various converters in the AC stage? Please specify.

6) Why three-phase bridgeless boost rectifier and three-phase Vienna rectifier are especially competitive for fast charging? You mention the power loss. I am confused because actually the use of diode usually deteriorates the efficiency while there are many diodes in the Vienna rectifier.

7) In recent years, charging safety has been quite a hotspot in the field of WPT, including overtemperature issues and the intrusion of metal objects. The power converter is used to control the power level or even directly shut down the charging system when thermal risks emerge.

8) You can consider simplifying the description of the working principle of DC-DC converters, and inverters. Or else you can detail the mainstream only. In the current version, it seems you are writing a book.

Author Response

(The authors gave the same response as above.)
